# Multidrug-Resistant Sepsis: A Critical Healthcare Challenge

**DOI:** 10.3390/antibiotics13010046

**Published:** 2024-01-04

**Authors:** Nishitha R. Kumar, Tejashree A. Balraj, Swetha N. Kempegowda, Akila Prashant

**Affiliations:** 1Department of Biochemistry, JSS Medical College and Hospital, JSS Academy of Higher Education & Research, Mysuru 570004, India; nishitharkumar@jssuni.edu.in (N.R.K.); swethank@jssuni.edu.in (S.N.K.); 2Department of Microbiology, JSS Medical College and Hospital, JSS Academy of Higher Education & Research, Mysuru 570004, India; tejashreea@jssuni.edu.in; 3Department of Medical Genetics, JSS Medical College and Hospital, JSS Academy of Higher Education & Research, Mysuru 570004, India

**Keywords:** sepsis, drug resistance, microbial, critical illness, mortality, healthcare costs

## Abstract

Sepsis globally accounts for an alarming annual toll of 48.9 million cases, resulting in 11 million deaths, and inflicts an economic burden of approximately USD 38 billion on the United States healthcare system. The rise of multidrug-resistant organisms (MDROs) has elevated the urgency surrounding the management of multidrug-resistant (MDR) sepsis, evolving into a critical global health concern. This review aims to provide a comprehensive overview of the current epidemiology of (MDR) sepsis and its associated healthcare challenges, particularly in critically ill hospitalized patients. Highlighted findings demonstrated the complex nature of (MDR) sepsis pathophysiology and the resulting immune responses, which significantly hinder sepsis treatment. Studies also revealed that aging, antibiotic overuse or abuse, inadequate empiric antibiotic therapy, and underlying comorbidities contribute significantly to recurrent sepsis, thereby leading to septic shock, multi-organ failure, and ultimately immune paralysis, which all contribute to high mortality rates among sepsis patients. Moreover, studies confirmed a correlation between elevated readmission rates and an increased risk of cognitive and organ dysfunction among sepsis patients, amplifying hospital-associated costs. To mitigate the impact of sepsis burden, researchers have directed their efforts towards innovative diagnostic methods like point-of-care testing (POCT) devices for rapid, accurate, and particularly bedside detection of sepsis; however, these methods are currently limited to detecting only a few resistance biomarkers, thus warranting further exploration. Numerous interventions have also been introduced to treat MDR sepsis, including combination therapy with antibiotics from two different classes and precision therapy, which involves personalized treatment strategies tailored to individual needs. Finally, addressing MDR-associated healthcare challenges at regional levels based on local pathogen resistance patterns emerges as a critical strategy for effective sepsis treatment and minimizing adverse effects.

## 1. Background

Sepsis is a critical medical condition associated with significant biological and chemical abnormalities that pose a high death rate. Unlike superficial and confined infections, sepsis is a complex disturbance of the delicate immunologic equilibrium between inflammatory and anti-inflammatory responses. This interaction demonstrates the fragile connection between the immune system and the clinical signs of sepsis. Over the past few decades, a comprehensive definition of “sepsis” has continuously evolved and improved [1]. Significantly, the current definition of sepsis (Sepsis-3) was proposed by the Third International Consensus, which defined it as “organ dysfunction caused by a dysregulated host response to infection”. This description is the first to stress the vital function played by the natural and acquired immune system response at the onset of a medical illness [2].

During the initial stages of sepsis, the immune system mediates the activation of pro- and anti-inflammatory cytokines, pathogen-related molecules, and mediators, leading to the initiation of the complement cascade and coagulation [3]. For instance, numerous endogenous host-derived signals like damage-associated molecular patterns (DAMPs) or exogenous stimulations like pathogen-derived molecular patterns (PAMPs), such as DNA fragments, lipids, exotoxins, and endotoxins, are the starting signals for sepsis. These molecules interact with toll-like receptors (TLRs) present on the surface of antigen-presenting cells (APCs) and monocytes, leading to the expression of genes associated with pro-inflammatory interleukins (IL, IL-1, IL-6, IL-8, IL-12, and IL-18), tumor necrosis factor-alpha (TNF-α), and interferons (IFNs like IFN-y) and anti-inflammatory (IL-10) pathways and acquired immunity [4,5]; these processes are usually observed during the initial stages of sepsis [6,7,8]. This upregulated inflammation progresses to concomitant immunosuppression, leading to progressive tissue damage, multi-organ failure, increased immune cell apoptosis, and T cell exhaustion, which all together result in “immunoparalysis”, thereby making sepsis patients prone to opportunistic and nosocomial infection [6,9]. A signal transduction caused by PAMPs- and DAMPs-mediated activation of monocytes and APCs causes the translocation of nuclear factor-kappa-light-chain-enhancer of activated B cells (NF-κΒ) into the cell nuclei. However, in short, the overall impact of the dysregulated immune response, whether hyper- or hypo-responsiveness, on the individual’s immunological response is highly personalized, leading to significant challenges in diagnosis [1].

Sepsis is a worldwide public health concern characterized by high rates of morbidity and mortality and a significant financial burden [10,11]. For instance, Rudd and coworkers recently revealed the alarming worldwide estimations of sepsis, as 48.9 million cases of sepsis were reported in 2017, with 11 million deaths attributable to sepsis [11]. In 2011, sepsis substantially burdened healthcare facilities in the United States with USD 20 billion in annual costs [12]. Additionally, numerous indirect expenses might dramatically impact the quality of life of patients with sepsis. For instance, older patients with sepsis may experience long-term severe health issues, such as cognitive impairment and functional disability [13]. Furthermore, a study on the sepsis burden in the Indian intensive care unit (ICU) revealed that the elderly population is more prone to sepsis due to multiple comorbidities caused by compromised immunity. The study found that 132 out of 387 patients with sepsis had septic shock, with the lungs (45.5%) being the most common site of infection. The mortality rate was 60.7% and 78.9% in old and very old patients, compared to a 45.6% mortality rate observed in younger adults [14]. Similarly, another study identifying sepsis burden in Malaysian ICUs revealed that aging was significantly associated with a 30-day mortality rate among elderly sepsis patients (particularly patients aged ≥65 years), with a high 30-day mortality rate (28.9%) among elderly sepsis patients [15].

Like acute myocardial infarction and cerebrovascular stroke, sepsis is a critical and persistent chronological condition. In the case of sepsis, early and correct usage of antimicrobial drugs is of utmost significance within the first hour of detection, concurrently with organ support. If the microbial pathogen emerges as an MDR, including the methicillin-resistant *Staphylococcus aureus* (MRSA), carbapenem-resistant *Enterobacteriaceae* (CRE), and MDR *Pseudomonas aeruginosa*, the therapeutic efficacy of currently available antimicrobial drugs is compromised, which hinders treatment success. Additionally, multidrug resistance poses a substantial risk of developing numerous sepsis-related adverse effects [16], necessitating prompt administration of the most appropriate antimicrobial therapy. However, while antibiotic resistance in bacteria is continuously growing globally, it poses a critical challenge to treating clinical infectious diseases, particularly those leading to life-threatening sepsis, septic shock, and multi-organ failure [17]. Additionally, as bacteria evolve, new mechanisms of resistance are emerging regularly and spreading worldwide, which are restricting current treatment options and making it challenging to treat prevalent infectious diseases [18]. Despite the persistent need for new antimicrobial drugs, major pharmaceutical industries have withdrawn from this field due to the rising costs of clinical trials, demanding approval criteria, and a general lack of economic viability [19,20]. This has widened the gap between the urgent public health need for effective antibiotics and the diminishing prospects of developing new antibacterial medications, resulting in a concerning situation [19].

Most patients with sepsis are given empirical antibiotic treatment without a prior confirmed diagnosis. This may raise the likelihood of developing multidrug resistance, accompanied by significant ecological adverse effects. Moreover, sepsis patients receive higher initial doses of empirical antimicrobial therapy regardless of organ failure, which may increase the synthesis of circulating pro-inflammatory and anti-inflammatory mediators, negatively impacting their overall health and well-being [21]. Additionally, the widespread misuse of antibiotics contributes significantly to increased mortality rates [22] and the surge in antimicrobial resistance (AMR). This misuse jeopardizes individual health and overburdens national healthcare systems financially [23]. One major contributor to antibiotic overuse is the unethical sale of antibiotics without proper prescriptions or diagnostic tests [24]. Similarly, self-medication practices often driven by economic constraints result in an incomplete antibiotic course, which promotes antibiotic resistance development due to suboptimal dosing [25]. Additionally, economic incentives for vendors to promote antibiotic sales make changing such practices challenging [26]. Furthermore, we are only now starting to understand the implications of antibiotic restrictions on outcomes and costs. We are hindered by the absence of universal ethical guidelines and comprehensive data on outcomes. Additionally, the concept of “best” and “effective” therapy varies significantly among groups, which makes the decision to select antibiotics difficult. Moreover, suboptimal antibiotic therapy cannot eradicate the infectious agent from the body, exposing affected individuals to the risk of adverse outcomes and wider antimicrobial resistance. Therefore, rational antibiotic usage primarily relies on identifying patients who, in fact, require treatment or optimizing treatment for a faster recovery [27]. In this respect, this review aims to comprehensively analyze the current burden of sepsis, the factors responsible for its development and increasing severity, and sepsis-associated healthcare challenges to reduce sepsis risk and improve MDR sepsis therapy, particularly in critically ill hospitalized patients.

## 2. Epidemiology and Burden of MDR Sepsis

Sepsis is a worldwide severe health issue. Septic shock is a subclass of sepsis distinguished by metabolic, cellular, and circulatory defects that increase mortality risk among sepsis patients. Due to increased prevalence and pathobiological, molecular, genomic, and medical complications, sepsis and septic shock pose a growing worldwide burden and a formidable challenge for emergency doctors [28]. Since the first consensus definition (Sepsis-1) of sepsis in 1991, the occurrence and prevalence of sepsis and septic shock have steadily increased, reaching about 49 million confirmed cases with 11 million confirmed sepsis-related mortalities worldwide in 2017 [29,30]. According to a 2016 systematic review conducted in well-developed countries, over 30 million hospital-treated sepsis cases were reported annually worldwide, and 5.3 million individuals died from sepsis [31]. Sepsis is also vital in the ICU, affecting around 30% of ICU patients, with significant regional differences [32]. A Chinese study reporting national incidence and mortality of hospitalized sepsis revealed an annual increase in hospitalized sepsis from 328.25 to 421.85 cases per 100,000 during 2017–2019. In light of these findings, the World Health Organization (WHO) confirmed sepsis as a worldwide health priority [30].

Incidence and fatality rates of sepsis vary significantly, with the most significant burden in Oceania, sub-Saharan Africa, and the South, Southeast, and East Asian regions. An Indian study (2007) identifying the epidemiology of sepsis identified 176 out of 230 cases of systemic inflammatory response syndrome (SIRS) caused by sepsis in patients in intensive therapy units (ITUs). The mean age of patients was 54.9 years, and 67% were male. Patients with severe sepsis had significantly high ITU mortality, hospital mortality, and 28-day mortality, which were 54.1%, 59.3%, and 57.6%, respectively. Additionally, the percentage of cases with infection being the primary cause of hospital admission was 89.8% [32]. Another Indian study conducted in 2016 identified the clinical microbiological profile of elderly sepsis patients. It revealed that 28.75% of cases were blood culture positive, of which 51.7% had Gram-negative infection and 48.30% had Gram-positive infection. Similarly, *Staphylococcus aureus* (49 patients) and *Escherichia coli* (36 patients) were the most prevalent pathogens isolated from sepsis patients. Subsequently, a 2017 study conducted in India identifying the sepsis burden in the adult population demonstrated that 282 of the total patients (4711) admitted to the hospital had severe sepsis, with 63.6%, followed by 62.8% and 56% hospital mortality, 28-day mortality, and ICU mortality, respectively [33]. The respiratory tract was the predominant site of infection among sepsis patients. Similarly, Gram-negative bacteria were the dominant cause of sepsis, with *Acinetobacter baumanni* being the most isolated pathogen. Additionally, researchers also found a significantly high mortality rate for sepsis patients, which was 85% [34]. A most recent study (2023) determining the clinical and demographic profile of elderly patients admitted to medical ICUs at a Tertiary Care Center demonstrated that sepsis was the most common cause of death among elderly patients. Moreover, bloodstream infections with Gram-negative pathogens were more prevalent than those caused by Gram-positive pathogens [35,36].

Besides India, a retrospective study by the National Mortality Surveillance System (NMSS) reported approximately one million sepsis-related fatalities in China [37]. Another Chinese study reported an estimated incidence of 328.25 cases per 100,000 populace in 2017 [38], slightly less than the previously reported incidence rate of 352.10 in the Western Pacific region. Still, it is significantly less than the incidence rate of 415.13 cases per 100,000 population in the Pan-American region [39]. Similarly, around 85% of cases and deaths occurred globally due to sepsis in low- and middle-income countries [11]. Moreover, sepsis can afflict people of any age or gender, and considerable differences exist in the burden of the illness. A three-year study from 2017 to 2019 found that sepsis afflicted the elderly population over 65 with a 57.5% incidence rate, followed by children under ten with a 20% incidence rate [38]. Similarly, in 2017, the global age-standardized incidence of sepsis was higher among females (716.5 cases per 100,000 population) than males (642.8 cases per 100,000 population) [11].

Numerous studies have found a correlation between the frequency and incidence of MDR sepsis and hospital stay within the last 90 days, a history of stroke, aging, and infection with MDR organisms (MDRO). These observations may be explained by the growing incidence of MDROs in hospital wards caused by the widespread antibiotic usage and transmission between healthcare staff and patients [40,41,42,43]. ESBL-producing *Enterobacteriaceae* appear to be the most common (9.7%) among all MDROs, with ESBL-producing *E. coli* and *Klebsiella pneumoniae* accounting for 35% of all *E. coli* and *K. pneumoniae* isolates [44]. The percentage of ESBL production among *Enterobacteriaceae* varies from nation to nation; however, it is on the rise throughout Europe, with Italy having one of the highest prevalences of ESBL-producing *Enterobacteriaceae* [45]. Another study found that being hospitalized within the past 90 days is a particular risk factor for ESBL *Enterobacteriaceae*. This finding demonstrates the significance of contact with the healthcare setting, necessitating the empiric administration of carbapenems to sepsis patients who have this risk factor [46]. However, with the emergence of CRE, treating sepsis patients has become a formidable challenge for physicians [16]. Similarly, stroke is another risk factor linked to the emergence of ESBL-positive bacteria, which may be attributed to extended hospital stays, nursing home stays, and indwelling invasive devices like gastrostomies, bladder catheters, and nasogastric tubes. Given the frequency of ESBL *Enterobacteriaceae* and if an infection with MDROs is suspected (for example, previous hospitalization), an adequate administration of selective antibiotic therapy can be considered for ESBL+ pathogens while awaiting culture results [46].

Sepsis also poses a significant economic burden on healthcare systems. The annual healthcare costs of sepsis in the United States were USD 20 billion in 2011 [12] and USD 24 billion in 2013–2014 [47], which were increased to USD 27 billion in 2019. Overall, sepsis costs the US healthcare system over USD 38 billion annually, making it the most expensive illness linked to hospitalization [48]. In India, an estimated sepsis cost per patient was USD 55 in 2005 [49], while a 2008 study proposed a projected estimate of USD 53 million for the Indian healthcare system in 2012 [50]. Similarly, before the COVID-19 pandemic, the annual costs of sepsis were about USD 1.3 billion per year in Ontario and Canada [51]. According to a nationwide study conducted in Japan, the adjusted annual gross medical cost of sepsis rose from USD 3.04 billion to USD 4.38 billion during the study period, which was linked to an increasing number of patients with sepsis (indicating 67,318 cases in 2010 to 233,825 in 2017). Another study discovered that shorter hospital stays were related to lower medical expenses [52]. These escalating healthcare-associated expenses have been attributed to prolonged hospital stays, expensive medications, and, regrettably, restricted access to treatment for sepsis patients, contributing to an alarming number of misdiagnosed sepsis-related fatalities. For instance, recent research estimated that sepsis affected 48.9 million people, and around 11 million people died globally in 2017, making sepsis responsible for approximately 20% of all global deaths [11]. Moreover, an eight-year Japanese study found that sepsis caused 18.9% of in-hospital mortality [52].

## 3. Pathogenesis and Mechanisms of Drug Resistance

In an ecological environment, bacteria are believed to strive for resources for existence, equipping various microbes with the complex chemical compounds yielded through metabolic activity that can inhibit or kill other microbes [53]. For instance, Penicillins and Cephalosporins are metabolic products of *Penicillium* and *Cephalosporium* species. A study identifying antimicrobial resistance (AMR) in archaea demonstrated that 30,000-year-old archaea were resistant to aminoglycoside (streptomycin) and β-lactam antibiotics (penicillin) [54]. Similarly, bacteria have evolved with time and developed resistance to antimicrobial drugs, a self-defense mechanism achieved through natural selection. Many antibiotic resistance mechanisms within bacterial metabolic pathways have additional functions to perform. For instance, the efflux pump that transfers particular antibiotics outside the bacterial cell membrane may also export toxic compounds like heavy metal ions to protect the bacterial cell [55]. Other antibiotic resistance mechanisms in bacteria involve adapting a latent state, structural and morphological changes, reduced permeability of bacterial cell walls and cell membranes, decreasing drug uptake, inactivating drugs, regulation of metabolism, target site modification, secreting target-protecting proteins, initiation of self-repair systems, and biofilm production, which all collectively constitute the defense system of bacteria against antibiotics [56,57]. Hence, rapidly spreading AMR across microbial populations cannot be caused by a single factor; instead, it involves multiple complex mechanisms.

Additionally, there are a few difficult-to-treat AMR pathogens categorized under the well-known abbreviation “ESKAPE”, which include *Enterococcus faecium*, *S. aureus*, *A. baumannii*, *K. pneumonia*, *Enterobacter* species, and *P. aeruginosa* [58]. As mentioned previously, all these pathogens have different mechanisms of resistance and thus cause varied degrees of infection. For example, *A. baumannii*, which causes hospital-acquired AMR infections, confers resistance to antibiotics by producing β-lactamases (all four classes: A–D) to degrade beta-lactam antibiotics, activating drug efflux pumps, producing modified porins to reduce drug permeability through bacterial outer membranes, and altering drug targeting sites [59]. Similarly, *P. aeruginosa* causes both acute and chronic hospital-acquired and severe respiratory infections. Like *A. baumannii*, *P. aeruginosa* can produce all four classes (A–D) of β-lactamases. Moreover, this pathogen can confer resistance through gene mutation, resulting in overexpression of *AmpC* β-lactamases. It can produce transferable aminoglycoside modifying enzymes (AMEs), which reduce the binding affinity of aminoglycosides to their target site in the bacterial cell [60]. Additionally, *S. aureus*, which causes mild and severe life-threatening skin and soft tissue infections, pleuropulmonary, bacterial endocarditis, and device-related infections, has decades of AMR history [61] attributed to the presence of penicillin-binding proteins (PBP and PBP2a) and genes, including *mecA*, *mecC*, *VanA*, *gyrA*, *gyrB*, and *erm* (*ermA*, *ermB*, *ermC*, and *ermF*) [62,63].

Besides AMR mechanisms, complex immune reactions involving the production and utilization of pro- and anti-inflammatory molecules, although aimed at protecting organisms from internal and external threats, lead to the excessive production of these inflammatory molecules. This, in turn, results in the rapid and simultaneous display of immune activation and immunosuppression signs in sepsis patients [64], as illustrated in Figure 1.

These concomitant secretions result in immunological paralysis, a significant reason for high mortality rates in patients who experience septic shock caused by MDR pathogens [65]. This could be attributed to previous exposure to an initial inadequate antimicrobial therapy, which cannot treat the infection; instead, it can affect the host defense system and may lead to altered immune function. Indeed, inadequate antimicrobial therapy can have detrimental ecological effects on the microenvironment as it can cause superinfection with MDR pathogens [66]. Similarly, weeks or months of continuous immune activation against pathogens, as in the case of sepsis patients, may lead to a chronic state, impairing the ability of cells to recognize antigens and creating a microenvironment where cells of innate and acquired immunity (neutrophils, macrophages, monocytes, T-cells, and B-cells) receive numerous stimuli that devastatingly affect their activity. The overall performance of receptors located on the cell surface and within the cell, which play a crucial role in the detection of microbial substances and internal warning signals, is crucially affected [67]. This concept is represented in Figure 2.

In the clinical management of sepsis, physicians strive to offer effective empirical antimicrobial treatment for hospitalized patients with sepsis, sometimes restoring to prescribing antibiotics without precise diagnostic confirmation. Unfortunately, while intended to save lives, this practice comes at the expense of potentially prescribing unnecessary antibiotics. This excessive treatment is associated with the emergence of MDR bacteria. Moreover, many patients use antibiotics without any prescription, whereas others take excessive doses of prescribed antibiotics, contributing to antibiotic resistance in bacteria. The higher incidence of multidrug resistance in sepsis patients could also be attributed to multiple patient-specific factors, including older age, comorbidities, immunosuppression or excessive use of immunosuppressive drugs, chemotherapy for cancer patients, and living in countries with lower and middle-income economies with deprived healthcare infrastructure and inaccessibility to healthcare facilities [68]. Polypharmacy, which involves the concurrent use of five or more drugs, is another significant factor contributing to the emergence of MDR bacteria in sepsis cases. Polypharmacy is often associated with the natural aging process, which, due to simultaneous biological and pathological changes, elevates the risk of multimorbidity and the necessity for multiple concurrent medications [69]. Since AMR in bacteria and fungi is complex and rooted in millions of years of evolution, these microorganisms have adopted different strategies to withstand antimicrobials, survive, and reproduce.

Consequently, most bacteria carry natural resistance to one or even multiple antibiotics. Contrarily, many bacteria can alter their antibiotic-targeting sites and become antibiotic-resistant. Similarly, self-medication is frequently non-specific to the target disease; hence, it may occasionally result in resistance development in opportunistic pathogens [70].

## 4. Common Pathogens Involved in MDR Sepsis

Sepsis can result in septic shock, multiple organ dysfunction, and ultimately death if it cannot be diagnosed timely and managed adequately. Sepsis can be infectious and caused by various microorganisms, including bacteria, viruses, and fungi. Bacteria are the most prevalent etiological pathogens, and *Streptococcus pneumoniae*, *S. aureus*, *E. coli*, *Hemophilus influenzae*, *Salmonella* spp., and *Neisseria meningitidis* are some of the most common bacterial pathogens involved in sepsis or sepsis-related comorbidities [71]. Fungi are responsible for around 15% of all infections, with aggressive fungal infections being the primary cause of sepsis, particularly in patients with immunosuppression or severe illnesses. For example, Candida species are the most prominent cause of fungal sepsis, responsible for around 5% of all sepsis cases. Invasive Candida infections are linked with a significantly increased sepsis-associated mortality risk. Various studies have linked inadequate antifungal therapy with higher mortality rates in patients with candidemia (a bloodstream infection-BSI caused by Candida species) or septic shock attributed to Candida [72]. Additionally, sepsis and septic shock indicators can be the lethal recurrent outcomes of infections caused by seasonal or periodic influenza, dengue viruses, and highly contagious pathogens of community health significance. Notable examples include swine and avian influenza viruses, the Middle East respiratory syndrome-related [MERS] coronavirus, the severe acute respiratory syndrome-related (SARS) coronavirus, and, most lately, the Ebola and yellow fever viruses [71].

Furthermore, anyone suffering from a severe infection, damage, or chronic disease can progress to sepsis; however, specific populations are more likely to develop the condition, including the elderly, pregnant or recently pregnant women, newborns, hospitalized patients, ICU patients, immunocompromised patients, and patients suffering from comorbidities or chronic medical conditions (like kidney disease or cirrhosis) [68]. Some other studies have also confirmed that populations of underdeveloped countries, females, and older people, particularly those with comorbidities, are high-risk populations [11,60]. Similarly, sepsis, which may be acquired in healthcare settings, is among the most common adverse events during medical care establishment. This condition affects hundreds of millions of individuals worldwide each year. Infections contracted in healthcare settings and frequently brought on by MDR bacteria are described in Table 1, which can rapidly deteriorate the clinical condition of patients. This is the reason behind the higher risk of hospital-associated mortality among sepsis patients infected with MDR pathogens [73]. Several studies evaluated a relationship between gender, infection, and risk of sepsis and found that male patients with respiratory infections have higher chances of developing sepsis than females (36% versus 29%). Contrarily, female patients with genitourinary infection are more prone to develop sepsis than males (35% versus 27%) [74]. Accordingly, BSI caused by *P. aeruginosa* and *S. aureus* is more prevalent in males than females [75]. Conversely, approximately 60% of BSIs with *E. coli* occur in females, consistent with the higher risk of females developing sepsis due to urinary tract infections [76]. Similarly, various published manuscripts have confirmed that male patients with candidemia have a higher risk of developing sepsis than females [77,78].

Additionally, some studies have identified a relationship between various factors affecting mortality rates among sepsis patients, including age, gender, comorbidities, disease severity [79], and the early initiation and appropriateness of antimicrobial and non-antimicrobial therapy [80]. One study identified a higher mortality rate of sepsis among individuals in the age group 15–50 years than older patients (58.5% versus 39.1%). The same study also revealed that most factors affecting mortality rates in sepsis patients are uncontrolled. They further found that 21.05% of sepsis patients were discharged from hospitals on medical advice. In general, the relationship between age and mortality among sepsis patients was controversial in this study [81]. Conversely, a study conducted by Carbajal-Guerrero and colleagues revealed that older patients (>65 years) with sepsis had a higher risk of comorbidities compared to the younger patients, and these comorbidities were found to be a potential factor contributing to the high mortality rate among the elderly [82]. Furthermore, the effect of gender on sepsis is still under debate among researchers. One study identified a higher incidence of sepsis among males than in females [83]. Another study that investigated the effect of gender on the survival of sepsis patients [83,84] showed that survival was better in females [85]. These differences in mortality rates for sepsis between male and female patients can be attributed to differences in their immune responses. For instance, estrogen production is higher in female patients than in males, which positively influences immune activity. This is because increasing body mass index and age in females increase the production of estrogen by elevating aromatase activity in adipose tissues, and high estrogen provides better protection to female patients with sepsis through immune activation [86].

## 5. Diagnostic Challenges and Innovations

Diagnosing sepsis at an early stage and promptly initiating treatment are essential for enhancing clinical outcomes and reducing the death rate of sepsis. Until a suitable alternative test is available, pathogen detection through conventional blood culturing has traditionally been the accepted method for diagnosing sepsis, as shown in Table 2. However, routine blood culturing takes 2–3 days to identify bacteria and even more time to test for antibiotic sensitivity, which is deemed inadequate in the case of sepsis. In such conditions, each hour of delay in treatment worsens patient conditions and increases morbidity and mortality [87]. Much research has been conducted on identifying the importance of blood culture for sepsis patients. One meta-analysis comprising 22,655 individuals with sepsis and septic shock from seven studies revealed only a positive blood culture result for 40.1% of patients [88]. Another study identified only 10–15% of positive blood culture results in neonates with sepsis [89]. Studies have confirmed multiple factors contributing to this poor diagnosis. For instance, most sepsis patients whose blood samples were taken for blood culturing had non-infectious inflammatory conditions caused by inflammatory, neurological, or metabolic disorders [90]. Conversely, sepsis patients with probably infectious inflammatory conditions receive antibiotics even before their sepsis worsens or before blood culturing, resulting in the inability of culture techniques to diagnose pathogens. Cheng and colleagues confirmed this phenomenon by demonstrating a 12% absolute difference in the count of positive blood culture outcomes before and after antimicrobial testing [91], which decreases the probability of detecting pathogens [92]. Finally, several microbial pathogens, such as fungi, bacteria, and some viruses, are undetectable through the traditional culturing approach and require alternative indicators for detection, including urinary antigens and non-specific markers for fungal presence. However, this may become increasingly challenging due to the rising incidence of sepsis caused by unusual pathogens [93].

These comparisons outline the key differences between the two approaches used in diagnosing antimicrobial resistance, highlighting the strengths and weaknesses of each method. Clinical diagnostic challenges and the need for immediate diagnosis and treatment have led to a dependence on identifying biomarkers in the blood, including procalcitonin (PCT), C-reactive protein (CRP), and white blood cell (WBC) count. Indeed, the early consensus definition of sepsis, Sepsis-1, incorporated a decreased (<4 × 10^9^/L) and an increased (>1.2 × 10^10^/L) WBC count into the criteria for systemic inflammatory response syndrome (SIRS) [94]. However, predicting infection in sepsis patients through serum biomarkers is debated due to the lack of sensitivity and specificity of many serological tests. A retrospective cohort study by Marik and Stephenson found a very poor predictive value (as low as 0.52 AUROC, an area under the receiver operating characteristic) of the WBC count for bacteremia in patients suspected of sepsis [95]. Similarly, Siegel and colleagues found a normal WBC count in 52% of patients with confirmed blood culture results showing bacteremia [96]. A meta-analysis study found that WBC count had minimal diagnostic significance in serious infections, with a negative probability ratio as low as 0.61 [97]. Similar results were obtained for CRP [98]. In contrast to the WBC count, the ratio of the neutrophil-to-lymphocyte count has constantly been found to be a far more accurate biomarker of physiological strain than absolute neutrophil or WBC counts [99]. An increase in neutrophil count and a decrease in lymphocyte count are frequently observed in systemic illnesses like sepsis, which may be attributed to the endogenous actions of hormones like cortisol and catecholamines. Moreover, sepsis prompts the migration of lymphocytes to inflammatory tissues, while increased lymphocyte apoptosis causes an increase in the ratio of neutrophils to lymphocytes [100]. A prospective study by Ljungström and a group comprising 1572 patients revealed a higher ratio of neutrophil-to-lymphocyte count compared to PCT and CRP (AUROC 0.68 versus 0.64 versus 0.57) or diagnosing bacterial sepsis [101]. However, a recent study predicting disease severity in COVID-19 patients confirmed that any kind of severe physiological strain can result in a rise in the ratio of neutrophils to lymphocytes, irrespective of the sepsis [102]. Additionally, studies confirmed that the neutrophil-to-lymphocyte ratio was invariably elevated even in non-infectious sepsis, making it significantly less precise to diagnose sepsis in critical care patients [103].

Although CRP is a commonly used biomarker in critical illnesses, it is non-specific for bacterial infections; instead, CRP levels increased in most other causes of inflammation. A meta-analysis study evaluating the diagnostic performance of CRP in sepsis identified that CRP has a better-pooled sensitivity (80%) but only 61% specificity [104]. Studies have confirmed that CRP levels have a minimal association with the disease severity in sepsis, whereas they serve as the most commonly used biomarker for predicting the disease severity in patients with pancreatitis [105], with 100% and 81.4% sensitivity and specificity, respectively [106]. However, CRP cannot constantly differentiate sterile from infected pancreatic necrosis. Therefore, it is not a suggested biomarker to initiate antimicrobial therapy [107]. Similarly, the production of PCT is increased in response to sepsis [108], and it rises within 2–3 h of infection and gains a peak at 24 h, which is a much quicker rise than CRP (which reaches a peak at 72 h). A systematic review and meta-analysis conducted by Wacker and colleagues reported that PCT has an AUROC of 0.85; thus, it is an excellent biomarker for distinguishing sepsis from other non-inflammatory syndromes [109]. Another meta-analysis study comprising 12 articles found that PCT may exhibit limited effectiveness in differentiating viral and bacterial infections, with Kamat and group identifying a poor sensitivity of 55% and a moderate specificity of 76% [110]. Additionally, PCT has lower sensitivity and diagnostic AUROC to predict bacterial infection in individuals with autoimmune diseases [111], chronic renal failure [112], and immunosuppression [113].

Novel diagnostic approaches for pathogen detection can be helpful alternatives to conventional techniques. Surface-enhanced Raman spectroscopy (SERS) is becoming an increasingly significant method for detection due to its ability to amplify the Raman scattering of target particles on a superficial layer of metal-made or graphene-based surface [114]. Moreover, this method can easily detect label-free nucleic acids. Similarly, numerous studies have confirmed that matrix-assisted laser desorption ionization time-of-flight mass spectrometry (MALDI-TOF MS) is a rapid diagnostic method for accurate identification of various microscopic life, including yeast, bacteria, fungi, and even Nocardia and mycobacteria species within a very short time frame, thereby minimizing the amount of time needed for adequate and effective antimicrobial therapy in sepsis [115,116,117]. Studies also confirmed that MALDI-TOF MS significantly reduced the hospital stay of sepsis patients by 1.75 to 6 days [115,116] and enhanced overall survival by 4 to 9% [115,117], thereby highlighting the significance of early detection of pathogens. Unfortunately, MALDI-TOF MS cannot identify AMR mechanisms, and testing antibiotic susceptibility depends on conventional methods [118]. However, more sophisticated systems have been developed that use polymerase chain reaction (PCR) for microbial amplification before MS detection to rapidly identify clinically relevant bacterial and yeast species with a higher diagnostic strength than cultural techniques [119]. These systems can also detect microbial species that do not typically grow in blood cultures, including *Mycoplasma pneumoniae*, *Rickettsia typhi*, *Legionella pneumophila*, *Nocardia* spp., and various fungi [120]. Although these systems can rapidly and efficiently diagnose the microbiological cause of sepsis, only a few studies have confirmed their usefulness over conventional cultures [121]. Furthermore, these methods are currently limited to detecting only a few of the large diversity of antibiotic resistance markers, which are crucial for providing tailored treatment [122].

Much research has demonstrated that timely diagnosis of sepsis episodes and medical intervention improve clinical outcomes [123]. Many other studies have identified that timely antibiotic treatment yields a lesser impact than the patient control group, indicating the variability of the disease and the necessity for continued analysis and medication [124,125]. Therefore, the optimal point-of-care sensors make it possible to rapidly compile patient health data, increase healthcare coverage, and improve the efficiency of healthcare services while simultaneously reducing healthcare costs [126,127]. Furthermore, it is extensive and fast enough to provide researchers with sufficient information regarding pathogen and host–response virtually anywhere in a very short time, which enables the treatment of sepsis in two major streams: firstly, POCT-based devices can speed up the identification step where optimum care is delayed, thereby improving outcomes, and secondly, they can identify numerous things, including pathogens, cell-surface proteins, and plasma proteins, which are ascribed as representatives of the immune response of hosts and which, when coupled with complex data analytics, can assist in stratifying sepsis even at the patient bedside. This kind of information might accelerate the procedure for detecting patients who may benefit from supplementary therapy [123]. POCT may also see the evaluation of the development of various protein biomarkers (such as IL-6, IL-10, PCT, CRP, and TNF-α) linked with acute sepsis and septic shock in ICU patients and estimate the probability of all-cause mortality within 28 days [128], assisting in the decision-making process for the selection of antibiotics.

## 6. Clinical Management of MDR Sepsis

Despite substantial advancements in our knowledge of the pathophysiology of sepsis, numerous clinical trials have been unsuccessful in identifying novel therapies that can alter the course of the disease [129,130]. Recognizing sepsis as a medical emergency is essential since, in the absence of definitive treatment, therapeutic interventions involve timely management of infection and organ support [131]. The 2016 Surviving Sepsis Campaign (SCC) guidelines strongly advise the prompt administration of intravenous broad-spectrum antibiotics, ideally within an hour following sepsis detection [132]. Several publications on sepsis and septic shock have found that delayed antibiotic administration is linked with adverse outcomes [133,134,135]. Beyond their apparent advantages, broad-spectrum antibiotics can cause substantial damage, such as antibiotic-associated adverse effects and potentially fatal AMR-related consequences [136,137]. Infections with MDRO have significantly increased worldwide, restricting our therapeutic options. The growing AMR is estimated to be responsible for approximately 10 million deaths each year by 2050. Therefore, treating patients with sepsis and septic shock by augmenting antimicrobial efficacy and avoiding the emergence of MDR strains is one of the primary concerns. Regarding this, antimicrobial stewardship (AS) is an important strategy for sepsis care since it focuses on multi-professional teamwork [for example, microbiologists, infectious disease specialists, and pharmacists] with appropriate, adequate, and optimized antimicrobial therapy [138].

Various studies have confirmed improved survival rates in sepsis patients with early and suitable antimicrobial administration and efficient source control [139], as validated by the inclusion of similar recommendations in 2016 SSC guidelines for early delivery of appropriate broad-spectrum antimicrobial drugs within one hour of hospital admission in patients afflicted by sepsis and septic shock [28,132]. Moreover, administering empiric antibiotic therapy directed at the most likely pathogens involved in infectious sepsis is crucial to improving patient outcomes. Numerous published manuscripts have discussed the adverse impact and consequences of inadequate empiric therapy in sepsis patients [138,140,141,142,143,144]. Notably, prescribing ineffective empiric therapy is prevalent in ICUs, occurring in 10–40% of sepsis cases, which varies depending on the frequency of MDR pathogens [144,145]. Recent studies have found that the patient group with higher disease severity scores is most likely to benefit from appropriate antibiotic treatment. In contrast, ineffective empiric antimicrobial therapy was linked with a 5-fold decrease in the survival of over 5000 individuals suffering from septic shock [133]. Another prospective study has found a significantly increased mortality rate among patients with septic shock and an average of three organ dysfunctions [143]. An appropriate empiric antimicrobial therapy means prescribing drugs that cover almost all potential pathogens responsible for the suspected infection. To achieve this, certain pathogen- and patient-related factors must be considered [138,146], including weight, age, allergies, comorbidities, chronic organ dysfunction, immunosuppressive therapy, and previous antibiotic or infection history. The risk of MDR pathogens should also be considered, including lengthy hospital stays, previous hospital admissions, the presence of invasive medical devices, and prior encounters with MDR pathogens [138]. Several investigations into the detrimental consequences and outcomes of delayed antimicrobial provision in patients with sepsis have concluded similar results [147,148,149]. These studies have confirmed that appropriate antibiotic therapy significantly decreased the mortality rate when it was given within ≤1 h [33], whereas each hour of delay in the treatment increased mortality [150] and dropped the overall survival rate by an average of 7.6% [87]. Besides delayed antibiotic administration, lengthy hospital stays [149,151], acute renal [152] and lung [153] diseases, and worsening organ dysfunction [154] have also been found to be common factors associated with increased mortality in sepsis patients.

Compared to these findings, various studies were unsuccessful in determining the usefulness of timely antimicrobial therapy [155,156,157]. A meta-analysis comprising over 16,000 individuals with sepsis and septic shock from 11 studies identified an insignificant difference between antibiotic administration (within 3 h) and mortality rate [158]. Another meta-analysis study comprising 11 studies on sepsis patients identified a 33% reduction in mortality among patients receiving early empiric antibiotic therapy (≤1 h) compared to those with delayed antibiotic administration (>1 h) [159]. A recent systematic review concluded that the mortality rate significantly decreased in patients with septic shock receiving early and adequate empiric antibiotic therapy [142]. Despite inconsistent outcomes, there is substantial agreement among international specialists on the need for prompt antimicrobial therapy in patients suffering from sepsis and septic shock, and novel ideas have recently been offered. A “door-to-needle” duration of 60 min has been advocated for antibiotic delivery, which indicates global concerns about launching a time window for successful therapy after sepsis detection [136]. Nonetheless, ensuring a competent application of institutional standards for antibiotic administration within 1 h after presentation remains difficult.

Given the rapidly growing prevalence of MDR infections, combined antibiotic therapy is commonly advised to warrant a larger antimicrobial spectrum and appropriate empiric coverage. The combined therapy is described as using antibiotics from two separate classes that have activity against a single infection, primarily to speed pathogen elimination and increase the susceptibility of pathogens to treatment [160]. To ensure the likelihood of having at least one active antibiotic against the possible pathogen involved, the Infectious Diseases Society of America (IDSA) endorses using two active medicines against Gram-negative bacilli for empiric treatment of septic shock [161]. Recognizing the need to encourage antibiotic judiciousness, the IDSA formed a committee to explore suggestions for prudent antibiotic usage in treating sepsis. The experts accepted ten antibiotic class combinations out of a total of 21. Concerns about rising resistance and proper pathogen coverage were stated as factors for selecting such combinations. The use of any combination involving macrolides or ciprofloxacin and specific pairings of aztreonam with cephalosporins and aminoglycosides with intravenous clindamycin were prohibited [162].

Studies on combination therapy have yielded conflicting findings, and there is a scarcity of well-powered randomized controlled trials examining this particular issue. Numerous observational studies, however, demonstrated that combination therapy outperformed monotherapy in individuals suffering from sepsis and septic shock [163,164]. For instance, a meta-regression analysis found a link between combination therapy and a high survival rate among severely ill sepsis patients with a higher mortality risk. Unexpectedly, this meta-analysis identified higher mortality among the patient group with a low risk of death [165]. Similar findings were reported in other studies where researchers linked higher mortality with nephrotoxic side effects leading to renal failure [166]. Based on these inconsistent findings, some specialists advocate employing a pair of antibiotics for the initial treatment of patients with septic shock and suspected MDR pathogen infections. Even with negative culture results, treatment can be cut down to personalized therapy at the minimum acceptable time after microbiological isolation or a satisfactory clinical response [167]. To assess the effectiveness of different antibiotic combinations, well-powered randomized controlled trials examining multiple antibiotic combinations in different situations should be conducted [168]. Additionally, individualized therapies tailored to patients’ unique conditions, like diabetes, renal or hepatic failure, or immunosuppression, can yield favorable results instead of applying an uniform approach.

As sepsis is frequently accompanied by organ dysfunction, supportive care and management of organ dysfunction are critical in sepsis treatment to reduce complications and improve patient outcomes. Hemodynamic support and mechanical ventilation are the two fundamental pillars of supportive care. Hemodynamic support entails maintaining proper tissue perfusion and oxygen supply, fluid resuscitation to restore blood pressure, and adequate organ perfusion. Vasopressor medications may also be required to treat refractory hypotension and to sustain cardiac output. Similarly, mechanical ventilation techniques, such as low tidal volume ventilation and prone posture, benefit sepsis patients with acute respiratory distress syndrome induced by sepsis. Furthermore, renal and liver function should be constantly monitored to maintain optimal fluid and electrolyte balance and the fine balance of acids and bases. Some patients may need hemodialysis as a renal replacement therapy to prevent damage to other bodily organs caused by fluid imbalance and the presence of creatinine and urea in the blood, which hinder sepsis treatment [168,169,170].

## 7. Impact of MDR Sepsis on Critical Care

Studies have confirmed that sepsis and septic shock are highly prevalent among critically ill patients, which essentially require early and appropriate empiric antibiotic therapy within the first hour to manage these situations effectively [171]. However, MDR sepsis presents formidable challenges within ICUs, significantly impacting patients’ well-being and straining healthcare resources. The complex nature of MDR microorganisms reduces antimicrobial treatment efficacy, often causing treatment failures and lengthy hospital stays. These MDR pathogens raise concerns about possible horizontal transmission within ICUs, highlighting the vital need for consistent infection prevention and control policies. Similarly, resource-restricted ICUs often lack essential equipment, laboratory assistance, and qualified physicians and nursing teams. Therefore, sepsis management guidelines in resource-limited ICUs, formulated by the Global Intensive Care Working Group of the European Society of Intensive Care Medicine (ESICM) [172], often differ in various aspects from the SSC recommendations, which were established in well-developed countries [173]. Notable instances include the meticulous management of glucose levels in the blood using insulin, a safe approach with consistent and accurate monitoring of blood glucose but risky when the effects of insulin on the blood are rarely or inadequately assessed. Furthermore, conventional culturing techniques cannot detect infectious sepsis due to empiric antibiotic administration to patients or take around 48–72 h to yield results. Therefore, early and precise identification of MDR pathogens is vital to support better infection control strategies [171].

Multiple infections, including ventilator-associated pneumonia (VAP) and hospital-acquired pneumonia (HAP), are widely prevalent in ICU settings and account for over half of all antibiotics provided in critical care situations. Despite attempts to enhance timely detection and therapy, the morbidity and mortality of sepsis and septic shock remain high, especially in patients with MDR sepsis [174]. Physicians in the ICU continue to have difficulty diagnosing VAP and HAP at the bedside. Routine CXR is no longer advised for ICU patients to evaluate disease progression and its response to treatment; instead, it is advisable to consider lung ultrasonography as a valuable diagnostic tool for VAP and HAP, especially when paired with the medical data of patients [171]. Previous studies have confirmed favorable effects and outcomes of β-lactam or β-lactamase inhibitors against VAP and HAP, especially for various Gram-negative bacilli that pose a significant concern in ICU settings. The β-lactam antibiotics are widely used in ICUs and are one of the safest antibiotics; however, they also have side effects. For example, neurotoxic symptoms have been identified in 10–15% of patients admitted to hospital ICUs. Similarly, there was an increased incidence of renal failure observed in ICU patients when they were administered β-lactam antibiotics in combination with nephrotoxic medicines like vancomycin [175].

Implementing strategies for controlling and preventing infection, including prudent antibiotic stewardship, strict adherence to hand hygiene protocols, comprehensive environmental disinfection regimens, and timely detection of MDR microorganisms, is critical to restrain the transmission of AMR pathogens. This necessitates a united effort and collaboration among healthcare practitioners, effective monitoring systems, and knowledgeable antimicrobial management teams to mount a staunch defense against the growing danger of MDR sepsis in critical care settings [176].

## 8. Frequency and Causes of Readmission in Sepsis Patients

Despite recent advancements in the medical field, the mortality rates associated with sepsis are significantly high, affecting almost 42% of sepsis patients [31]. However, alarmingly, even patients who survive are not immune to the effects of sepsis, as nearly one-third of sepsis survivors were readmitted within 180 days. Readmissions following sepsis-related hospital stays are frequent and expensive, with severe physical and financial implications. The relationship between surviving sepsis and subsequent readmissions is a relatively new area of research, with prior studies focusing solely on short-term and immediate outcomes. Consequently, we could only find a few studies for comparison, all from well-developed countries. The national study by Norman and group [177] found a 30-day readmission rate of 28% in the United States. Another study comprising patients from 21 community-based hospitals [178] found a readmission rate of 17.9%. Similarly, a 90-day readmission rate ranged between 30 and 42% [179]. Research conducted by Goodwin and colleagues on 43,452 sepsis survivors admitted to non-governmental hospitals in California, New York, and Florida found a significantly high 180-day readmission rate of 48% [180]. This high readmission rate may be attributed to a greater risk of depression [180], sleep deprivation, encephalopathy [177], mental illnesses, and cognitive and organ failure, all ultimately leading to death among sepsis survivors, as identified by various studies [181]. Besides high morbidity, the high readmission rate of sepsis survivors also comes with a significant financial burden, as recent research quoted an annual cost of over USD 38 billion spent on sepsis in the United States. On average, a solitary readmission may result in expenses ranging from USD 25,000 to USD 30,000. These horrifying figures can be attributed to the fact that sepsis is generally treated in the ICUs, which is extremely costly due to the cost of lengthy hospital stays, medications, laboratory tests, use of medical equipment, invasive devices, procedures, nursing staff, and taxes [182,183].

The financial burden on sepsis patients is exacerbated in developing countries, including India and Pakistan, where patients typically have to pay for healthcare-associated expenses out-of-pocket. Additionally, patients do not have medical insurance or loan facilities. A brief look at the per capita figures in developing countries puts these findings into proper perspective. With a per capita income of USD 1500 in a developing country compared to a substantial USD 53,000 in the United States, it is easy to assume how a single readmission could be overwhelming for patients and their families. Most households in developing countries have only one worker; therefore, an illness leading to prolonged hospitalization for that individual could be disastrous for the entire family. Moreover, the majority of employees live paycheck-to-paycheck and have few savings or investments. There is no choice for sick leave, and each day spent in the hospital results in no revenue for that day. Furthermore, it would not be easy to find a suitable substitute for the primary wage earner due to cultural factors in most patriarchal families. Consequently, families find themselves compelled to liquidate all of their assets or borrow money from relatives and friends, which might take years to repay. Other family members commonly offer nursing care in the home, resulting in reduced focus on childcare and diminished earning potential [184].

## 9. Preventive Measures and Infection Control

The Antimicrobial Stewardship Program (ASP) is a multifaceted, collaborative approach that engages various healthcare professionals, including clinicians, microbiologists, pharmacists, and nursing staff, to enhance treatment outcomes and prevention by minimizing AMR among microbial pathogens [185]. The ASP is one of three important principles of a comprehensive strategy for strengthening healthcare systems. Although infection prevention and control (IPC) and medicine followed by patient safety are the other two principles of ASP, ASP cannot be successful without including IPC [186] because healthcare epidemiologists and infection preventionists play a pivotal role in the implementation and success of ASP [187]. Notably, following the WHO essential medications list “AwaRe16” classification [Access, Watch, and Reserve], optimizing antibiotic usage, and surveillance are important aspects of ASP that are directly linked with reduced AMR [188]. This multifaceted strategy eliminates the need for antimicrobial therapy by preventing infection transmission, which reduces the emergence of resistance. The Centers for Disease Control and Prevention (CDC) developed seven fundamentals for ASP implementation in 2019. Notably, leadership and accountability are the first two concepts or principles responsible for the program’s goals and outcomes, followed by education and local antibiogram deployment. The latter two are administrative components, which are based on the idea that common infections receive appropriate empiric therapy. Prescription preauthorization and resistance surveillance performed by pharmacists and laboratorians, respectively, are the two actionable tasks where the necessary interventions can be carried out as mandated by institutional standards and policies [189].

The WHO has designated AMR as a global threat because it is a well-established fact that threatens public health and national security [190]. Therefore, the association between healthcare providers (HCPs) and public health organizations is critical. It makes it easier to develop prevention initiatives, promote education, and conduct surveillance, all aimed at slowing down the spread of AMR [191]. Patients who exhibit resistance to currently available antimicrobials force physicians to employ reserved antibiotics like carbapenems and polymyxins. These reserved antibiotics are expensive, may not be readily accessible in some countries, and may have potentially unintended consequences [for example, colistin administration is linked with acute kidney injury [192]. Currently, healthcare professionals are facing a worldwide challenge of MDR-ESKAPE pathogens, which are infamously branded as “bugs without borders” [193,194]. These are nosocomial pathogens with the ability to escape the biocidal effect of antimicrobials [195]. Hence, tackling AMR is a crucial aspect of ensuring safe and successful healthcare delivery, as highlighted by the implementation of ASP [196]. Since its start, ASP has been extremely effective in reducing antibiotic usage. Notably, the four Ds, which are the key facets of ideal antimicrobial therapy, encompass selecting the right drug, dose, de-escalation to pathogen-directed therapy, and the right duration of therapy and infection control. These are the guiding principles of ASP [195]. These approaches align closely with public health objectives and encompass the promotion of ASP by monitoring, ensuring data transparency, developing infrastructure, and increasing patient and healthcare professional knowledge and awareness [191].

The lack of novel antimicrobials necessitates the preservation of existing ones. To ensure the judicious use of novel antimicrobials, the Infectious Diseases Society of America [IDSA] and other public health bodies recommend the implementation of ASP to preserve the efficacy of these medications [83]. Moreover, a set of systematic ASP initiatives have been introduced globally in clinical settings to lessen selected pressures that favor highly resistant organisms [197]. The ASP is critical in preventing the AMR spread [198]; however, a meta-analysis study found significant variability in the included studies, and collaborations between the IPC department and the ASP team were found to be more effective in limiting the AMR spread. Nonetheless, it is recommended that all fundamental features, including education programs and antimicrobial limitation through prospective audits and feedback, be employed in conjunction to improve outcomes. Notably, ASP efforts may not produce results without hospital leadership commitment. ASP has been found effective in reducing AMR and hospital costs in various regions worldwide, and a few of the safety measures and prevention controls are illustrated in Figure 3 [199,200].

Everyday self-care routines that incorporate cleansing and sanitizing both your body and hands are paramount to maintaining good health. Regular sterilization of surfaces prone to high contact is also significant in curbing the spread of harmful microorganisms. Face masks should be worn consistently, particularly when maintaining safe distances from others is difficult. Self-medication is a practice to avoid, especially in cases where the correct dosage and timing of intake are not known. To prevent potential contamination risks, hospital waste should be correctly deposited into the designated trash receptacles. Travel plans should be put on hold when one is unwell as a preventive measure against spreading the disease.

MDR microbial pathogens cause a significant proportion of infections in ICUs, with around 23,000 deaths annually in healthcare settings alone in the USA [201]. Besides host susceptibility, the complexity and logistics of critical care medications put patients at risk of contracting infectious pathogens. Invasive procedures and implantable devices, which are frequently used to provide supportive care to critically ill patients, also serve as entry points for pathogens. Similarly, the concurrent involvement of numerous medical team members and the utilization of numerous patient care devices for lifesaving critical care treatments may increase the chances of infection transmission from staff or fomites to patients. Generally, infection control precautions may not be prioritized in emergency conditions like sepsis and cardiac arrest, in which even seconds matter. Pathogenic microorganisms in the ICU are more prevalent on or in the human body [skin, respiratory epithelium, and gastrointestinal tract] and in the hospital environment and serve as transmission reservoirs. Additionally, antibiotics, chemotherapy, or acquiring nosocomial pathogens, among other things, might disrupt a patient’s flora. Therefore, patients colonized with resistant bacteria can serve as potential reservoirs for the transmission and spread of infection. The proportion of patients in a given unit colonized with resistant bacteria, or colonization pressure, is an independent risk factor for transmission [202,203]. Moreover, person-to-person transmission of resistant pathogens mainly occurs through contaminated patient care equipment, the hands of healthcare providers, and contaminated surfaces.

A recent study found environmental contamination with MDROs in 40% of patient rooms in the hospital, including Vancomycin-resistant Enterobacterales [VRE] [204]. Studies also found the viability of difficult-to-treat MDROs like MRSA, VRE, and *A. baumannii* on fomites in the hospital environment, including dry surfaces, steel, and plastic materials. Other pathogens of high concern were also found prevalent under dry conditions, like carbapenemase-producing Enterobacteriaceae, including *bla_KPC_*-carrying *Klebsiella pneumoniae* [205]. Studies have shown the high efficacy of approved hospital disinfectants against these pathogens. Using disposable patient care equipment, especially those known to be used for patients harboring MDROs, has been reported to minimize the risk of cross-transmission. Additionally, sharing items, including cooling blankets, blood pressure measuring devices, and portable radiology cassettes, should be thoroughly disinfected. Other items, like fabric privacy curtains, should be replaced with disposable curtains [206]. Numerous studies have documented the benefits of supplementary methods of disinfection, like hydrogen peroxide vapors and ultraviolet lights, to reduce the burden of bacterial pathogens and their spores. Hydrogen peroxide is effective in decontaminating hospital wards experiencing outbreaks [207] or environments where high-concern pathogens are present [208]. Contrarily, other studies have identified that although ultraviolet lights are less labor-intensive, less time-consuming, and do not require technical expertise for operation, they are less effective in eliminating all pathogens. Similarly, some pathogens can reside in damp environments and may form biofilms from which they can be transmitted to patients. For example, waterborne bacteria, including *Stenotrophomonas*, *Pseudomonas*, *Aeromonas*, and *Sphingomonas*, can colonize plumbing fittings such as sink drains, faucets, and aerators.

Preventing transmission via contaminated plumbing is a significant concern in hospital infection control and is currently being researched [209]. Some basic methods include ensuring that hospital water has an adequate amount of free chlorine, choosing sinks with low-splash designs, and keeping patient care items away from handwashing sinks, where they could be polluted by pathogen-contaminated drain splash-back. Plumbing fixtures may require disassembly, special cleaning and disinfection measures, or even replacement in an epidemic environment in which plumbing fittings are implicated [210]. Therefore, ASP, hand hygiene, and adequate disinfection of hospital surfaces and equipment are essential in preventing the spread of MDROs. Further, hospital administrations comprising infection control specialists, microbiologists, and critical care experts should collaboratively constitute policies and procedures for infection control in critical care units and emergency rooms, the necessary training and education of ICU staff for infection control, and other relevant outcome measures. Additionally, infection control protocols and procedures must be followed by having adequate nursing personnel, setting up infrastructure like handwashing stations, and providing hospital supplies like masks, gloves, and alcohol-based hand gels.

Furthermore, vaccines are commonly administered as a preventative measure and are applicable before the bacteria grow and spread following the initial infection (during low pathogen burden) and before various tissues and organs are affected. This significantly lowers the probability of mutations that confer resistance arising and spreading. Antibiotics often only have one mode of action or one target, like the cell wall of bacteria or bacterial translation machinery. This is because antibiotics are designed to be highly specific in killing pathogens. Bacteria can naturally resist antibiotics or acquire or develop this resistance over time (like avoiding access to antibiotic targets, drug efflux, modifications of drug targeting sites, or even inactivation of the antibiotics themselves). Therefore, changes in the drug target site caused by a single mutation render the antibiotic useless. Additionally, the selective pressure from antibiotic usage encourages the development of drug-resistant clones. Conversely, vaccines reduce the likelihood of resistant clones being selected for further development since they have a preventative effect. Moreover, because vaccines frequently target several antigens or various epitopes of the same antigen, for instance, polyclonal antibodies, the development of vaccine-evasion variations would require many mutations that would each have an impact on a distinct epitope, making the emergence of resistance in bacteria challenging [211].

## 10. Global Efforts and Collaborations

To underscore the serious threats posed by AMR, the CDC has published a study to characterize the important challenges associated with AMR and threat level classifications for MDROs [212]. The report classified pathogens into three distinct types: urgent, serious, and concerning. With ESKAPE pathogens being the most urgent threat to sepsis patients, policymakers and stakeholders have initiated numerous programs in this area. For instance, the National Action Plan for Combating Antibiotic-Resistant Bacteria (CARB) was launched to confront the escalating challenge of AMR through a well-coordinated and collaborative effort as part of the US government’s national response focused on addressing AMR. Five areas were focused on by the action plan, including (i) reducing and stopping the emergence of resistant bacteria, (ii) strengthening One Health monitoring efforts, (iii) promoting the development and use of rapid and novel diagnostics for detecting resistant organisms, (iv) expediting research for new antibiotics, alternative therapeutics, and vaccines, and (v) improving global collaboration [213]. Additionally, the WHO has approved an action plan focusing on AMR with five goals, including (1) increased awareness of AMR through efficient communication and education, (2) strengthened knowledge and evidence base for monitoring and scientific research, (3) decrease in the frequency of infection through infection prevention and hygiene measures, (4) optimization of the judicious use of antimicrobials in both human and animal health, and (5) creation of an economic rationale for long-term investment that considers the demands of all countries [214]. From a public health perspective, the CDC has led a multidimensional effort involving activities aimed at detecting and treating resistance on time and investing in prevention measures. The establishment of an Antibiotic Resistance Solutions Initiative, the Antibiotic Resistance Lab Network, and fundamental advice for ASP in various healthcare settings [215] are specific CDC activities.

The word “stewardship” was first coined in 1970, when an international initiative for optimal antibiotic administration, dosage, and duration was taken. In 2012, the Global Sepsis Alliance (GSA), an organization committed to decreasing the influence of sepsis and coordinating national and global initiatives against sepsis, introduced World Sepsis Day. Before that, numerous national public health organizations were unfamiliar with sepsis knowledge; even the Global Burden of Disease Report did not mention sepsis. Later, the White House issued the National Action Plan to demand the implementation of ASPs by 2020 in all hospitals providing acute care to patients. In this regard, by 2016, 64.2% of the critical care hospitals in the US had satisfied the essential criteria of the ASP proposed by the CDC [216]. The CDC has focused on ASP by releasing recommendations called “the Core Elements of Hospital Antibiotic Stewardship Programs”. The basic features are designed to help hospitals of all sizes and complexity confront the dangers of AMR while also promoting patient safety through the deployment of effective ASPs. The guidelines recognize the dynamic nature of ASP and the need for greater flexibility in project and program implementation. Key components include leadership dedication, responsibility, pharmacy knowledge and expertise, action, monitoring, reporting, and education [215]. The Centers for Medicare and Medicaid Services provided additional support and engagement in 2019 by mandating the establishment and advancement of an ASP as a prerequisite for participation for all acute care hospitals participating in Medicare and Medicaid programs [217]. Over the years, these policies and activities have aided in the formulation and execution of ASPs across various settings, encompassing both integrated and non-integrated healthcare systems. Indeed, forward-thinking healthcare systems have initiated efforts to encourage and subsidize ASPs. Similarly, Europe has taken numerous initiatives to implement ASPs at regional and national levels [218]. In this regard, ESGAP, the ESCMID Study Group on ASP, has played an especially prominent role in these activities.

Besides these initiatives, public awareness of the AMR problem is critical. A survey analysis using the Amazon Mechanical Turk Crowdsourcing platform to recruit respondents found that, despite a substantial majority of respondents (93%) agreeing that unsuitable antibiotic usage contributes to antibiotic resistance, 70% of the survey respondents expressed a neutral stance or disagreed with the assertion that antibiotic resistance is a problem [219]. Another poll found that 65% of the American populace perceives antibiotic resistance as a matter of public health concern, and 81% are concerned that diseases may become progressively more difficult to treat as a result of antibiotic resistance. An annual observance to raise awareness about AMR was held as part of the CDC’s initiatives to combat AMR and involve the public [220]. Increased public education regarding the substantial strain imposed by antibiotic-resistant infections on healthcare resources and the communal issues involved in a holistic approach to countering AMR will remain critical in the future. Regarding sepsis, the WHO took significant measures to address the pressing global health threat of sepsis, resulting in the publication of the WHO Secretariat Report and the adoption of Resolution WHA70.7 by the 70th World Health Assembly (WHA) in May 2017 on “Improving the prevention, diagnosis, and clinical management of sepsis”. The first progress report on implementing the resolution was issued in 2020 for WHA 73. Among the significant accomplishments were identifying sepsis treatment gaps and developing global guidelines for the clinical management of sepsis [221].

Sepsis Alliance, founded in 2007, is another prominent sepsis organization working in all 50 states of the US to “save lives and reduce suffering from sepsis.” Sepsis surveillance is dedicated to saving lives and improving suffering by enhancing sepsis awareness and treatment. Its goal is to make this world free of sepsis. Sepsis Alliance is also a proud co-founder of the GSA, founded in 2010, and currently represents over one million caregivers in over 70 countries [222]. GSA initiated World Sepsis Day [WSD] in 2012. Since then, events have occurred worldwide every September 13th to promote awareness about sepsis. Various events are also organized for medical personnel, including sports activities, pink picnics, photo exhibitions, dinners, grand galas, multiple possibilities for public gatherings, including hospital open houses and community healthcare events, and online campaigns, including the “World Sepsis Congress”, and movements across various social media websites like Facebook, Twitter, and WhatsApp [221]. Similarly, the International Surviving Sepsis Campaign (SSC) is a collaborative project of the Society of Critical Care Medicine (SCCM) and the European Society of Intensive Care Medicine (ESICM), which are dedicated to lowering morbidity and mortality occurring globally due to sepsis and septic shock. SCCM is also committed to enhancing the prognosis for sepsis survivors, particularly those with post-sepsis syndrome. The SSC campaign was initiated in 2002 during the annual meeting at ESICM and has established guidelines and bundles for managing sepsis [168].

Several challenges to lowering the massive global burden of sepsis include difficulties in identifying related morbidity and mortality, insufficient knowledge, poverty, health inequities, resource-limited public health, and a fragile acute healthcare delivery system. Context-specific solutions to this serious problem are essential due to considerable disparities in susceptible populations, the infecting microorganisms, and the healthcare ability to manage sepsis globally, particularly in low and middle-income countries (LMIC) [223]. The high variability of typical critical care syndromes, including sepsis, has hampered developments in finding therapy targets; consequently, the demands of severely ill patients in LMIC are frequently unmet, and some patients are even subjected to therapies that might be harmful. Given the substantial resource variance, it may be impossible to anticipate identical goals and worldwide agreement in all management areas. Therefore, regional critical care management teams nationwide must customize diagnostic and treatment methods for various problems in their respective environments. Similarly, investments in the acute care of sepsis patients should be appropriate and effective compared to expensive and technology-concentrated ones. Such assets can provide substantial returns across several clinical specialties and positively affect population health outcomes [224,225].

## 11. Discussion

Sepsis is a life-threatening emergency condition of global public health concern with substantial mortality and financial costs. Sepsis definition has significantly evolved in recent years, with the currently acceptable Sepsis-3 definition, which emphasizes the role of the immune system in sepsis development. This review comprehensively evaluated various research and reports on MDR-sepsis and its associated healthcare challenges. Epidemiological data from various studies highlighted a high prevalence and incidence of sepsis, with significant disparities at regional and global levels. One study reported 48.9 million cases of sepsis, with 11 million deaths occurring annually worldwide. Another study found that the annual healthcare costs of sepsis reached USD 38 billion alone in the USA. Studies conducted in India revealed a higher incidence of sepsis among elderly ICU patients and with Gram-negative bacterial pathogens, particularly *E. coli* and *A. baumanni* [34,35]; however, one Indian study also identified Gram-positive *S. aureus* as the prevalent cause of sepsis [34]. Similarly, studies identified a significant increase in the number of MDR sepsis among hospitalized neonates and the elderly, notably with Gram-negative bacterial pathogens. Studies have shown consistency regarding the causative pathogens of sepsis. For instance, most studies highlighted the presence of ESKAPE pathogens in infectious sepsis [226,227,228], whereas only a few studies identified *K. aerogenes* and *Enterobacter cloacae* as being responsible for sepsis [229].

The worldwide escalating incidence of sepsis and sepsis-associated healthcare costs may be attributed to the growing incidence of AMR in MDR pathogens, which significantly challenge sepsis treatment. Antibiotic resistance development in MDR pathogens, particularly against the commonly prescribed antimicrobials, results in substantial delays in providing effective antimicrobial treatment. These delays in treatment even worsen the health conditions of susceptible populations like children, the elderly, those with a previous history of infection, and patients with comorbidities. These delays are also correlated with increased mortality rates, prolonged hospital stays, and increased healthcare expenses. Consequently, in the face of MDR infections and delayed microbiological results, the widespread use of broad-spectrum antibiotics in empirical antimicrobial therapy has become a common practice. However, this reliance on broad-spectrum antibiotics further risks individual health by continuing the overuse and misuse of these drugs, consequently exacerbating the development of antimicrobial resistance [230].

Infections with *Enterobacteriaceae*, including *E. coli* and *K. pneumoniae*, are a significant concern in ICU patients with MDR sepsis. Studies have identified an almost 51% incidence of infection in ICU patients, with infection incidence density ranging from 13 to 20.3 episodes per thousand patient days [231]. A study conducted from June 2009 to December 2013 identified a 14.9% mortality rate among patients infected with *Enterobacteriaceae* bacteremia. The authors identified that increasing sepsis severity was significantly correlated with higher mortality, with 3.5%, 9.9%, and 28.6% mortality for sepsis, severe sepsis, and septic shock, respectively. They further identified that time to antimicrobial therapy was not significantly associated with mortality; however, prolonged ICU and hospital stays were found to be significantly associated with increased severity of sepsis, ultimately increasing the death rates among sepsis patients [232]. Furthermore, researchers from another study found that 48% of *Enterobacteriaceae*-infected patients developed recurrent infections within a 12-month follow-up period. Over half of these recurrent infections were caused by the same bacterial species and at the same culture site. Studies also identified that patients harboring MDR Gram-negative bacteria were independent predictors of subsequent mortality after discharge from the index hospitalization. Furthermore, researchers found that the chances of recurrent infection were high within the first three months of hospital discharge [233]. This is a significant finding, as it indicates a critical post-hospitalization period for monitoring and intervention. Therefore, timely and precise prognosis and outcomes for ICU sepsis patients with MDR *Enterobacteriaceae* infections are critical in managing MDR sepsis.

The AMR emergence in bacteria is a complex phenomenon attributed to diverse molecular mechanisms that rely on both the antimicrobial agent in question and the specific pathogen. These AMR mechanisms encompass a spectrum of genetic events, including constitutive or inducible expression of resistance genes and upregulation of these resistance genes [234]. Additionally, certain bacteria inherently possess resistance to specific types or entire classes of antimicrobial agents. Notably, in infectious diseases, bacteria can produce biofilms, which are implicated in over 65% of human infectious diseases [235]. Composed of structured entities like extracellular polymeric substances—polysaccharides, proteins, and extracellular DNA [236]—bacterial biofilms confer resistance through multiple mechanisms, including impeding the cell cycle, facilitating horizontal gene transfer, secreting enzymes that alter or bind antibiotics, and limiting antibiotic diffusion [237].

Studies have identified varied patterns of resistance among the pathogens involved in sepsis. As ESKAPE pathogens were found to be the major cause of MDR sepsis, pathogens have shown noticeable resistance to β-lactam antibiotics [58]. Most ESKAPE pathogens were found to be involved in the production of all four classes of β-lactamase enzymes [59]. Other pathogen resistance mechanisms involved in MDR sepsis were the production of AMEs by *P. aeruginosa* [60] and various resistance proteins and genes by *S. aureus*, including penicillin-binding proteins (PBP and PBP2a), *mecA*, *mecC*, *VanA*, *gyrA*, *gyrB*, and *erm* (*ermA*, *ermB*, *ermC,* and *ermF*) genes [62,63]. Additionally, studies have confirmed that most resistance cases were also attributable to patient-specific factors like older age, inadequate or excessive empiric antibiotic therapy, antibiotic usage without any prescription, immunosuppression [68], and in some cases, concomitant secretions of inflammatory and non-inflammatory molecules, leading to immunological paralysis [65]. Moreover, ESBL production is seen only in one-third of *E. coli* in early-onset sepsis, compared to a far higher ESBL production by *K. pneumonia* in late-onset sepsis [238]. Similarly, a Chinese study identified a significantly higher proportion of MDROs among patients with late-onset sepsis [239].

A comprehensive analysis of multiple AMR mechanisms reveals the diverse nature of antibiotic resistance patterns in sepsis-related pathogens. For instance, ESKAPE pathogens dominate in cases of MDR sepsis, which is attributed to their ability to produce multiple β-lactamase enzymes that target β-lactam antibiotics. Specific resistance mechanisms in *P. aeruginosa* and *S. aureus*, involving AMEs and varied resistance proteins and genes, highlight the complexity of resistance, which indicates that although sepsis can be treated through an appropriate antimicrobial regime, treating MDR sepsis is challenging due to the development of resistance in sepsis pathogens, which not only restricts the treatment options but also reduces the survival rate of patients, where each hour delay in treatment significantly increases the patient’s mortality rate. Moreover, patient-related factors like age, inappropriate antibiotic usage, immunosuppression, and the release of inflammatory molecules were also found to contribute significantly to the emergence of MDR sepsis. Variations in ESBL production between *E. coli* and *K. pneumoniae* in early and late-onset sepsis, along with a higher prevalence of MDROs in late-onset sepsis, highlight temporal and pathogen-specific resistance dynamics. These insights emphasize the multifaceted nature of antibiotic resistance in sepsis, demanding tailored treatment strategies that account for microbial and patient-specific complexities.

Timely diagnosis and management of sepsis are crucial for minimizing mortality rates. Previous studies have focused on conventional blood culturing techniques for appropriate empiric antibiotic therapy; however, diagnosing sepsis is still under debate among researchers due to the infectious as well as non-infectious nature of sepsis. For instance, antibiotic therapy can be applicable only to patients with infectious causes of sepsis. However, culturing techniques take a longer time to detect pathogens, thereby increasing the mortality rates of sepsis patients [90]. In this regard, various researchers mentioned CRP, WBC count, and PCT tests as significant biomarkers for diagnosing sepsis and initiating prompt antibiotic therapy [94]. Unfortunately, studies have found variable results in serological tests. They identified a lower to normal WBC count even in patients with culture-positive bacteremia [96]. Similarly, a higher neutrophil-to-lymphocyte count was considered a positive biomarker for infectious sepsis; however, many patients with even non-infectious sepsis were identified to have higher neutrophil-to-lymphocyte counts [102]. In addition to having low sensitivity and specificity, inconsistent findings were observed in the case of CRP [106] and PCT levels, making them less valuable for diagnosing sepsis [110]. Considering these issues, researchers have shifted their focus to identifying novel approaches for pathogen detection to diagnose sepsis. For instance, SERS [114] and MALDI-TOF MS techniques [115,116] have shown promising results in diagnosing sepsis. However, unfortunately, AMR mechanisms cannot be detected by MALDI-TOF MS [118], thus requiring an advanced system capable of identifying microbial AMR genes. In this regard, researchers have designed MALDI-TOF MS systems that incorporate PCR for microbial amplification before MS detection. These systems have a higher diagnostic potential than conventional culture techniques [119], and they can even detect microorganisms that do not normally grow in blood cultures, including *Mycoplasma pneumoniae*, *Rickettsia typhi*, *Legionella pneumophila*, *Nocardia* spp., and various fungi [120]. However, only a few studies have confirmed their usefulness over conventional cultures [121]. Therefore, further research is needed to develop technologically advanced systems to rapidly and accurately identify microbial genes associated with multidrug resistance in sepsis patients.

Additionally, the optimal POCT tests are rapid and extensive in providing sufficient information regarding pathogen and host–response virtually anywhere in a very short time [126,127]. Although these novel diagnostic techniques are promising, they have certain limitations. They can currently detect a limited number of resistance biomarkers. Hence, additional research is required to use these tests as a standard for diagnosing sepsis.

## 12. Strengths and Limitations

Focusing on MDR sepsis in relation to healthcare settings is one of the key strengths of this review. While previous reviews have examined aspects of MDR sepsis, none have focused on the increasing problem of MDROs or the healthcare burden of MDR sepsis. Another strength of this review was its rigorous approach to finding studies focusing on multiple aspects of MDR sepsis, from sepsis epidemiology and healthcare burden and cost to the evaluation of different pathogens involved and their resistance mechanisms. This review aimed to address a comprehensive range of pertinent topics related to MDR sepsis.

The type of evidence obtained determines the limitations of this review. These were geographically distributed and the result of various methodological approaches. None of the included studies utilized an in-depth qualitative approach to investigate the complete spectrum of factors that may affect the patient’s experience of care. Moreover, there was a lack of studies describing the significance of MDR sepsis and its impact on healthcare in the subcontinent region, particularly in India and Pakistan. Finally, only peer-reviewed and published research was included because it was deemed to be of the highest quality [237,238,239,240]. Gray literature was not included; this may be regarded as a further limitation because a deeper search in this direction may have generated additional material that could have contributed to this review and expanded its scope.

## 13. Future Directions in Research and Therapeutics

The field of sepsis care has seen a significant transformation over the past few decades, notably in therapeutic approaches. There have been various advancements, ranging from more precise therapies and the creation of new drugs to the discovery of novel alternative antibiotics. These recent advancements indicate an ever-changing landscape on the cusp of redefining sepsis care. Developing cutting-edge novel therapies and medicines is a step in the right direction, holding promising outcomes. Researchers have investigated various immunomodulatory drugs and targeted therapies to disrupt the complicated mechanisms that drive the advancement of sepsis [241]. These therapies have the potential to tip the scales in favor of patient recovery since they focus on reducing excessive inflammatory responses. However, as the case of drotrecogin-α demonstrates, turning scientific promise into clinical success is complicated and requires careful inspection [130].

The escalation of AMR necessitates the development of novel antibiotic resistance mitigation techniques. The failure of traditional therapy can be because the pathophysiology of sepsis is the consequence of a highly complex series of mechanisms in which a dysregulated host response produces cellular damage, tissue damage, and, eventually, organ failure. Therefore, to enhance the effectiveness of antibiotics, it is advisable to employ adjunct therapies that complement antibiotic treatment, such as improving supportive care, targeting bacterial virulence factors, and targeting host response factors. Supportive care involves employing oxygenation or ventilation strategies or optimizing fluid or vasopressor use based on patient-specific characteristics. Bacterial virulence factors can be targeted by using anti-endotoxin antibodies or endotoxin removal columns [242,243]. Hemadsorption methods, such as polymyxin B adsorption, are an example of an endotoxin removal column that has shown potential for filtering out endotoxins and creating new pathways to neutralize the detrimental effects of septic shock [244].

Similarly, host response factors can be targeted using anticoagulants or anti-cytokine drugs [243]. Contrarily, researchers have identified various other options that can serve as substitutes for antibiotic therapy. For instance, phage therapy may cause distinct forms of immunomodulation in successive phases of sepsis. Interestingly, the possible applicability of phages [and their enzymes, lysins] in treating sepsis has previously been supported by animal and clinical experiments [245]. However, these techniques require rigorous validation and incorporation into comprehensive care paradigms.

Immune-based therapies to alleviate the sepsis burden have consistently failed to improve patient outcomes. Recent advancements in immune medicine against cancer and the realization that extended immunosuppression in sepsis patients can leave them susceptible to secondary infection and mortality have prompted a renewal of sepsis immune therapy research. Earlier immune treatments were based on targeting a single mediator and were administered to varied patient groups with complicated and dynamic immune responses. In this regard, personalized immune therapy is on the rise due to advances in genomics, proteomics, metabolomics, and point-of-care technologies, together with an enhanced knowledge of sepsis pathophysiology [246]. During the past decade, changing preferences from immunosuppressive medications to immunostimulatory treatments have displayed promising effects in preclinical studies, case series, and small clinical investigations [247]. For instance, immunostimulatory agents such as interferon-gamma [IFNγ] and granulocyte–macrophage colony-stimulating factor (GM-CSF) have been the most widely researched in sepsis. These compounds exhibit robust potential for stimulating myeloid cell activity: they improve the antigen presentation capabilities by enhancing the monocyte human leukocyte antigen-DR [mHLA-DR] gene and the production of pro-inflammatory cytokines by monocytes [248]. A randomized controlled trial, guided by biomarkers that are associated with GM-CSF, compared its effects against a placebo and found it to be safe and effective in restoring monocytic immunocompetence. The treatment positively impacted sepsis patients, with preliminary results indicating that the mechanical ventilation time was reduced and disease severity declined swiftly in treated individuals [249]. The advent of precision medicine is ushering in an era of personalized therapies tailored to the individual [250]. Immune-based treatments, such as monoclonal and polyclonal antibodies and other immunomodulators tailored to the specific patient profile, represent a paradigm shift [251]. Utilizing the body’s natural defense, these treatments aim to retune immunological responses, minimizing the damage caused by sepsis. As we decipher the complex interplay between individual genetics, immunological state, and treatment results, the potential of precision medicine emerges as a beacon of hope. Compared to traditional immunosuppressive therapies, precision medicine is pivotal for the success of upcoming immunostimulatory drug studies. Consequently, it is crucial to identify individuals with a highly repressed or overactive immune system who are expected to benefit from immunosuppressive and immunostimulatory medication and to evaluate the immunological and therapeutic response accurately [252].

## 14. Conclusions

Sepsis is a complicated disruption of immunologic equilibrium, highlighting its complexity and the intricate link between immune function and clinical symptoms. The mortality, morbidity, and economic impact of sepsis are global concerns. Non-target-specific antibiotic therapy, misuse or abuse of antibiotic therapy, polypharmacy, and inadequate empiric antibiotic therapy may favor the emergence of MDROs, thereby having substantially adverse ecological side effects and economic burdens. That is why patients with infectious sepsis, particularly those harboring MDROs, have a higher risk of hospital-associated mortality. Antimicrobial resistance, including MDR pathogens, challenges treatment efficacy, increases the risk of adverse effects, and hampers treatment success. Antimicrobial resistance determines treatment ineffectiveness in clinical settings, leading to rapid advancement to sepsis and septic shock. Multidisciplinary strategies for timely diagnosis and application of appropriate antimicrobial treatment are critical in managing septic patients and limiting sepsis-related complications. Therefore, there is an urgent need for early administration of antimicrobials and organ support due to the time-dependent nature and severity of sepsis. Further, researchers should focus on developing diagnostic methods such as POCT that would detect sepsis early in the infection to avoid critical damage to organs and identifying better and more effective alternatives to antibiotics, such as phage therapy, immune-based therapies involving monoclonal and polyclonal antibodies, and precision medicine. Healthcare stakeholders must prioritize early and adequate administration of antimicrobials, preferably within the first hour of diagnosis, along with organ support. Public health organizations like the WHO collaborate with worldwide organizations and stakeholders to improve the treatment of sepsis and infection prevention control, including vaccinations, which should yield maximum outputs. With technological advancements, the POCT’s role in bedside detection of sepsis is substantially increasing. The use of nanoparticles and immune-based therapeutics, in combination with precision medicine, is an important field of research for healthcare providers, including physicians, pharmacists, and microbiologists. Furthermore, addressing the challenges associated with AMR is essential to ensure effective treatment and minimize adverse effects.

## Figures and Tables

**Figure 1 antibiotics-13-00046-f001:**
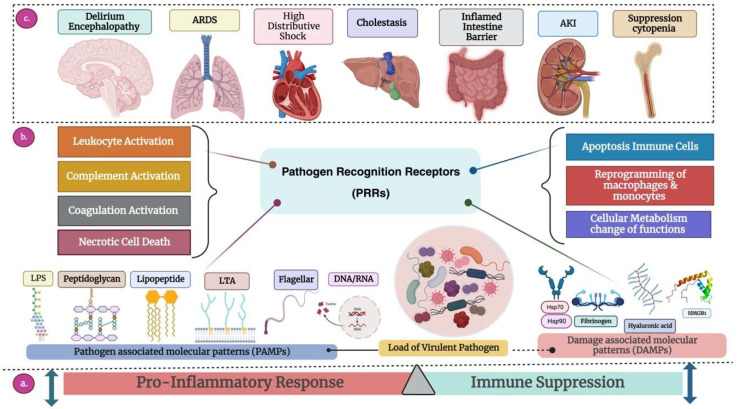
Overview of the pathogenesis of sepsis. (**a**) Immune Response: Sepsis occurs when the body responds to infection with an excessive immune system reaction, causing a disturbance in the usual equilibrium of the inflammatory response to maintain homeostasis. Activation of PRRs initiates both proinflammatory responses and immune suppression, ensuing hyperinflammation and immune suppression to the extent that is detrimental to the host. (**b**) Receptor Response: Once a pathogen successfully breaches the host’s mucosal barrier, it can induce sepsis, depending on its quantity and virulence. The host’s defense system identifies molecular components of invading pathogens (PAMPs) through specialized receptors called PRRs. This activation triggers the expression of target genes responsible for proinflammatory cytokines (resulting in leukocyte activation), inefficient utilization of the complement system, coagulation system activation, simultaneous downregulation of anticoagulant mechanisms, and necrotic cell death. This sets in motion a detrimental cycle, leading to the progression of sepsis, exacerbated by the release of endogenous molecules from injured cells (DAMPs or alarmins), further stimulating PRRs. Immune suppression manifests as extensive apoptosis, causing depletion of immune cells, reprogramming monocytes and macrophages into a state with reduced capacity to release proinflammatory cytokines, and an imbalance in cellular metabolic processes. (**c**) Organ Response: Organs respond to internal or external stimuli by initiating inflammation, undergoing changes in function, or activating compensatory mechanisms aimed at maintaining homeostasis and resolving disturbances. These responses are crucial for the body to cope with stress, injury, infection, or other challenges, ensuring proper functioning and survival. The main organs and their specific responses are described below. 1. Brain: (i) Delirium: Acute disturbance in attention and cognition, leading to confusion and altered perception. (ii) Encephalopathy: Brain dysfunction causing altered mental function, affecting cognition, consciousness, and behaviors. 2. Lungs: Acute Respiratory Distress Syndrome (ARDS) triggered by MDR bacteria is a severe and potentially life-threatening condition characterized by the rapid onset of widespread inflammation in the lungs. Infections, especially severe bacterial infections caused by multidrug-resistant bacteria, lead to direct lung injury, cytokine storms, secondary infections, and ventilator-associated pneumonia (VPA). 3. Heart: High distributive shock with MDR sepsis places immense strain on the heart due to systemic vasodilation and reduced blood flow, leading to compromised cardiac function and potential myocardial damage. The combination of multidrug-resistant sepsis and shock increases the risk of cardiac dysfunction, contributing to the severity of the condition and complicating treatment. 4. Liver: Cholestasis during MDR sepsis involves a disruption in bile flow due to both the effects of severe infection and potential liver dysfunction from multidrug-resistant bacteria. This combination worsens jaundice, impairs detoxification processes, and contributes to the systemic complications of sepsis. 5. Gastrointestinal tract: An inflamed intestine barrier exacerbated by multidrug-resistant bacterial infections leads to severe inflammation and compromised intestinal integrity, increasing the risk of bacterial translocation. This can result in the systemic dissemination of pathogens, exacerbating MDR sepsis. 6. Kidney: In MDR sepsis, acute kidney injury is a combination of sepsis-induced circulatory changes, and the potential nephrotoxicity of the pathogens contributes to kidney dysfunction, increasing the risk of severe complications and mortality. 7. Suppression cytopenia: During MDR sepsis, suppression cytopenia leads to a significant reduction in blood cell counts. The combination of multidrug-resistant pathogens and the immunosuppressive effect of sepsis increases the risk of complications, including compromised immunity and susceptibility to bleeding or infections. Abbreviation: ARDS, acute respiratory distress syndrome; AKI, acute kidney injury; DAMPs, danger-associated molecular patterns; DNA, deoxyribonucleic acid; HMGB1, high-mobility group box-1 protein; HSPs, heat shock proteins; LPS, lipopolysaccharide; LTA, lipoteichoic acid; PAMPs, pathogen-associated molecular patterns; PPRs, pattern recognition receptors; RNA, ribonucleic acid. The dashed lines depict the disrupted immune response triggered by infection, which makes the body unable to restore its equilibrium and causes harm to the organs. This culminates in a severe and life-threatening state known as sepsis.

**Figure 2 antibiotics-13-00046-f002:**
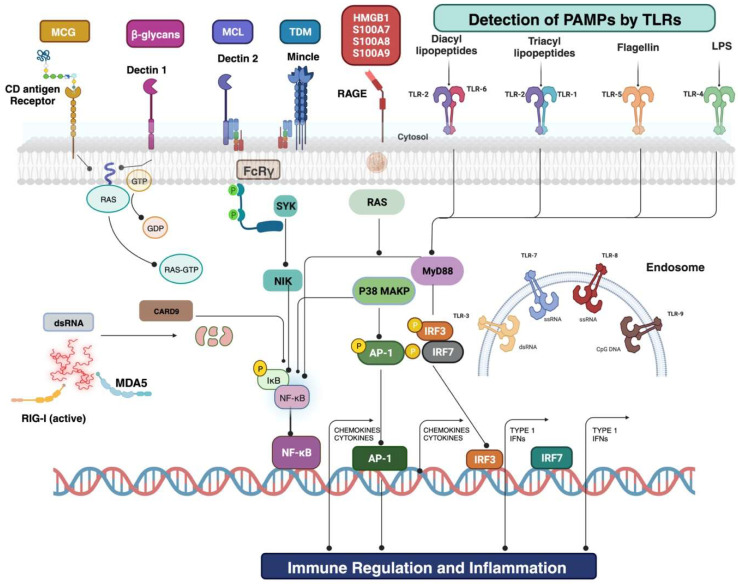
Cell surface and intercellular receptors amend for the recognition of PAMPs and DAMPs. The onset of sepsis is heralded by the host’s detection, prompting the activation of inflammatory signaling pathways. An extensive array of cellular and intracellular receptors is used to identify PAMPs or DAMPs. Examples include microbial and host-originated glycoproteins, lipoproteins, and nucleic acids. The corresponding PRRs encompass Toll-like receptors, dectin 1 (a member of the C-type lectin domain family 7), and dectin 2 (a member of the C-type lectin domain family 6). At least ten distinct TLRs have been identified, usually forming homodimers or heterodimers. Upon activation, these signaling pathways typically integrate into interferon regulatory factor signaling and nuclear factor-κΒ. IRF is in charge of type I interferon production. NF-κΒ and activator protein 1 signaling predominantly oversee the early activation of genes involved in inflammation, such as TNF and IL1, as well as those encoding for endothelial cell surface molecules. Among the other notable components within this sepsis-related network are caspase recruitment domain-containing protein 9, lipopolysaccharide, myeloid differentiation primary response protein 88, and stimulator of interferon genes protein. Loss of lymphocytes is directly immunosuppressive, contributing to the lymphopenia observed in patients. The genetic mutation or pharmacological intervention that decreases sepsis-induced apoptosis improves survival in severe sepsis. The degree of lymphocyte apoptosis in animal models of sepsis correlates with the severity of sepsis, and persistent lymphopenia predicts sepsis mortality. The next generation of treatments evaluated for suppressing immune function through interaction with sepsis includes therapies targeting lymphocytes and leukocytes. Abbreviations: CARD9, caspase recruitment domain-containing protein 9; dsDNA, double-stranded DNA; dsRNA, double-stranded RNA; FcRγ, Fcγ receptor; HMGB1, high-mobility group box 1; iE-DAP, d-glutamyl-meso-diaminopimelic acid; LGP2, laboratory of genetics and physiology 2; LPL, lipoprotein lipase; LPS, lipopolysaccharide; LY96, lymphocyte antigen 96; MAPK, Mitogen-activated protein kinase; MCG, mannose-containing glycoprotein; MDA5, melanoma differentiation-associated protein 5; DAMPs, damage-associated molecular patterns; MDP, muramyl dipeptide; MYD88, myeloid differentiation primary response 88; TLRs, Toll-like receptors; C-type lectin domain family 7 member A (dectin 1) and C-type lectin domain family 6; NIK, NF-κB-inducing kinase; NOD, nucleotide-binding oligomerization domain; RAF1, RAF proto-oncogene member A (dectin 2); RIG-I, retinoic acid-inducible gene 1 protein; ssRNA, single-stranded RNA; STING, stimulator of interferon genes; NF-κB, nuclear factor-κB; SYK, spleen tyrosine kinase; NF-κB and activator protein 1 (AP-1).

**Figure 3 antibiotics-13-00046-f003:**
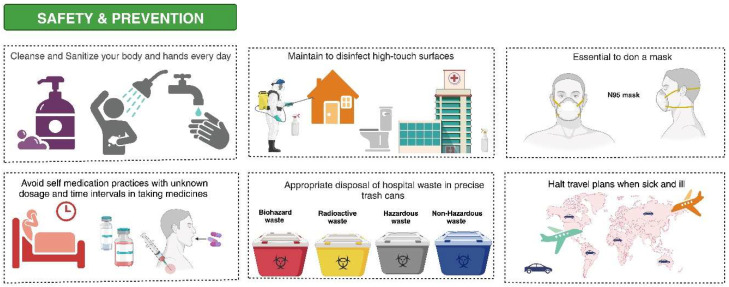
Prevention and control of the rise in multidrug-resistant microorganisms. Everyday self-care routines that involve cleaning and sanitizing your body and hands are paramount to maintaining good health. Regular sterilization of surfaces prone to high contact is also significant in curbing the spread of harmful microorganisms. Face masks should be worn consistently, particularly when maintaining safe distances from others is difficult. Self-medication is a practice to avoid, especially in cases where the correct dosage and timing of intake are not known. Hospital waste should be correctly deposited into the designated trash receptacles to prevent contamination risks. Travel plans should be put on hold when one is unwell as a preventive measure against spreading the disease.

**Table 1 antibiotics-13-00046-t001:** A systematic table covering reported mechanisms of multidrug-resistant bacteria in sepsis.

Gram-Positive Bacteria		
Bacterial Species	Mechanisms of Multidrug Resistance	Association with Sepsis
*Staphylococcus aureus* (including MRSA)	Altered penicillin-binding proteins (PBP2a)	Increased severity of infections, including skin and soft tissue infections, pneumonia, and bloodstream infections.
	Efflux pumps	MRSA is commonly associated with healthcare-associated infections.
	Biofilm formation	Virulence factors contribute to pathogenicity.
*Enterococcus faecium/faecalis* (including VRE)	Altered target site (D-Ala-D-Ala to D-Ala-D-Lac)	Frequent in healthcare-associated infections, especially in immunocompromised patients.
	Biofilm formation	High resistance to vancomycin, a crucial antibiotic.
**Gram-Negative Bacteria**		
*Escherichia coli* (Including ESBL-producing)	Production of extended-spectrum beta-lactamases	High resistance to beta-lactam antibiotics, leading to challenging treatment
	Plasmid-mediated resistance	Common in urinary tract, respiratory, and bloodstream infections.
	Porin mutations	Associated with nosocomial infections, which can progress to sepsis.
*Klebsiella pneumoniae* (Including CRE strains)	Production of carbapenemases	Limited treatment options due to resistance to broad-spectrum antibiotics.
	Plasmid-mediated resistance	High mortality rates associated with bloodstream infections.
	Reduced permeability of the outer membrane	Commonly found in healthcare settings.
*Acinetobacter baumannii*	Efflux pumps	Common cause of healthcare-associated infections, especially in ICUs.
	Biofilm formation	Associated with high mortality rates in bloodstream infections
	Intrinsic resistance mechanisms	Often involved in ventilator-associated pneumonia and septicemia.
*Pseudomonas aeruginosa*	Efflux pumps	Commonly implicated in hospital-acquired infections, including sepsis.
	Biofilm formation	Infections associated with a higher risk of treatment failure.

**Table 2 antibiotics-13-00046-t002:** A comparison table outlining the differences between conventional methods and molecular methods for diagnosing antimicrobial resistance (AMR).

Sl No	Aspect	Conventional Methods	Molecular Methods
1	Sample Type	Limited range of sample types	More adaptable to various sample types
2	Identification Speed	Relatively slow, it may take days to provide results.	Rapid results, often within hours.
3	Sensitivity and Specificity	It may have lower sensitivity and specificity.	Generally, higher sensitivity and specificity
4	Range of Pathogens Detected	Limited to certain pathogens (Genera of the Pathogen)	Broad range, capable of detecting various pathogens (exact Species of the Pathogen)
5	Type of Information	Phenotypic information (e.g., growth inhibition).	Genotypic information (specific genes or mutations).
6	Multiplexing Capability	Limited ability to test for multiple resistance genes	High multiplexing capability, detecting multiple targets in a single test
7	Equipment Required	Often requires specialized equipment and expertise	Requires specific equipment but can be more accessible
8	Ease of Use	It may require trained personnel and specialized equipment.	User-friendly protocols, less technical expertise needed
9	Accuracy	Subject to human handling error	Less prone to human error, higher accuracy
10	Resistance Detection Method	Culture-based methods, susceptibility testing	DNA sequencing, PCR, genotypic assays
11	Cost	Lower initial cost in some cases	Higher initial cost, but potentially cost-effective over time

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
