# Peer review of "Multidrug-Resistant Sepsis: A Critical Healthcare Challenge"

_antibiotics, 2024, doi:10.3390/antibiotics13010046_

Round 1

Reviewer 1 Report

Comments and Suggestions for Authors

I have reviewed the manuscript entitled “Multidrug-resistant sepsis: A critical healthcare challenge” submitted for possible publication in the journal “Antibiotics”. The authors have done great efforts in compiling all the relevant literature to address to problem. The title is fine and representing the content of manuscript. The manuscript structure needs extensive editing with minor English proofreading. The layout of figures needs to correct according to the journal requirements. There is more unnecessary information in the introduction and discussion section which should be removed. The manuscript can be proceeded further for possible publication after addressing some comments and suggestions. My specific comments are:

1.     Manuscript needs to be checked for formatting. The authors names need to be properly placed. The “and” should be moved before last author name.

2.     The bacterial names need to be checked again throughout the manuscript. e.g., page 4, kleb.

3.     The figure 1 need to be revised, captions need to be added in the figure legend. e.g., what mean by a, b, c etc. Figures needs more explanation in the legend.

4.     Figure 3 legend: replace “resistance” with “resistant”.

5.     The section “Future Directions in Research and Therapeutics” should be moved before the conclusion section.

6.     The authors are suggested to add some data related to sepsis in their country. The current measure to address the issue.

7.     The authors can provide a table of comparison for diagnostics of AMR earlier. E.g., conventional methods v/s molecular methods.

Comments on the Quality of English Language

Minor changes are required.

Author Response

  1. Manuscript needs to be checked for formatting. The authors names need to be properly placed. The “and” should be moved before last author name. - As per the suggestions received from the reviewer, the manuscript is thoroughly checked for formatting, and the author's names are placed properly.
  2. The bacterial names need to be checked again throughout the manuscript. e.g., page 4, kleb. - As suggested by the reviewer, the bacterial names are checked throughout the manuscript. 
  3. The figure 1 need to be revised, captions need to be added in the figure legend. e.g., what mean by a, b, c etc. Figures needs more explanation in the legend. - As suggested by the reviewer, the figure 1 is revised and caption is added. More explanation is added to the legend. 
  4. Figure 3 legend: replace “resistance” with “resistant”. - The changes have been made as suggested by the reviewer. 
  5. The section “Future Directions in Research and Therapeutics” should be moved before the conclusion section. - As suggested by the reviewer, the section “Future Directions in Research and Therapeutics” is moved before the conclusion section. 
  6. The authors are suggested to add some data related to sepsis in their country. The current measure to address the issue. - More data related to sepsis in India is added in the section "Epidemiology and Burden of Sepsis".
  7. The authors can provide a table of comparison for diagnostics of AMR earlier. E.g., conventional methods v/s molecular methods. As suggested by the reviewer, a table that compares conventional and molecular methods for the diagnosis of AMR is included in the revised manuscript. 

Reviewer 2 Report

Comments and Suggestions for Authors

Kumar et al.'s manuscript, titled " Multidrug-resistant sepsis: A critical healthcare challenge" presents a detailed examination of the antibiotic resistance mechanisms in sepsis. The study sheds light on the intricate mechanisms through which Multidrug-resistant bacteria role in sepsis. The manuscript by Kumar underscores an important and often underestimated aspect of Multidrug-resistant in sepsis. However, to enhance its suitability for publication in this journal, several crucial revisions and expansions are necessary:

           The introduction should be substantially revised and expanded to provide a more comprehensive and detailed overview of the subject matter, demonstrating a deeper understanding of the topic.

           The abstract should be more informative, effectively conveying the key points and findings of the study in a concise manner.

           The authors did not discuss the role of bacterial resistance in sepsis for a few specific bacteria. It is important to clarify why only these particular multidrug-resistant bacteria play a role in sepsis.

           Incorporate a systematic table to comprehensively cover reported mechanisms of multidrug-resistant bacteria in sepsis, providing a clear and organized presentation of the data.

           Include more data and surveillance reports on hospital sepsis to strengthen the study.

           The Discussion section requires expansion to offer a thorough interpretation of the findings, emphasizing their significance and potential implications in the context of antibiotic resistance.

           Prior to submission, the manuscript should undergo careful proofreading to ensure the proper usage and correctness of the English language throughout the document.

Comments on the Quality of English Language

Need to improve.

Author Response

  • The introduction should be substantially revised and expanded to provide a more comprehensive and detailed overview of the subject matter, demonstrating a deeper understanding of the topic. - As suggested by the reviewer, the introduction has been revised to provide a deeper understanding of the topic.
  • The abstract should be more informative, effectively conveying the key points and findings of the study in a concise manner. - Abstract has been made more informative as per the reviewers suggestions. 
  • The authors did not discuss the role of bacterial resistance in sepsis for a few specific bacteria. It is important to clarify why only these particular multidrug-resistant bacteria play a role in sepsis. - We have discussed those bacteria that show the highest prevalence of sepsis, however, the revised manuscript has a table mentioning the mechanisms of multidrug-resistant bacteria in sepsis.
  • Incorporate a systematic table to comprehensively cover reported mechanisms of multidrug-resistant bacteria in sepsis, providing a clear and organized presentation of the data. - As suggested by the reviewer, we have included the details in table 1
  • Include more data and surveillance reports on hospital sepsis to strengthen the study.- More data has been added in the revised manuscript, as suggested by the reviewer. 
  • The Discussion section requires expansion to offer a thorough interpretation of the findings, emphasizing their significance and potential implications in the context of antibiotic resistance. - We have expanded the discussion section and have emphasized the significance and implications of MDR sepsis
  • Prior to submission, the manuscript should undergo careful proofreading to ensure the proper usage and correctness of the English language throughout the document. - The English editing services of the journal office have been used to modify the revised manuscript. 

Reviewer 3 Report

Comments and Suggestions for Authors

In this review article, the authors reviewed the studies aimed at reducing the healthcare burden that multidrug-resistant (MDR) sepsis poses, particularly in critically ill hospitalized patients.

Comments

The reviewer has some concerns as follows:

1.     In the author list, it seems to be incomplete in the end: “…Akila Prashant a and c*”.

2.     The line numbers are lacking in this manuscript.

3.     In the Introduction section, the specific aims can be added.

4.     The format of this manuscript can be modified a little. Too many square brackets ([ ]) are used in the manuscript, which can easily be confused with cited references. It is recommended to use round brackets (( )) except for square brackets for cited references.

5.     In Figure 1, the presentation is confusing. The figure legend is too brief to understand the meaning of the figure. What are the meanings of (a), (b), and (c)? What does it mean for the dot-lines in the middle of the figure?

6.     The descriptions in the second line from the bottom, page 4 to line 27, page 5 are confusing. It should be re-organized and clearly explained.

7.     In Figure 2, the whole names for abbreviations are recommended to be described in the figure legend.

8.     In page 9, line 2, [> 12 × 109/L] can be changed to [> 1.2 × 1010/L].

9.     In the middle paragraph of page nine, what are the meanings for “sensitivity” 0.8 or 0.55 and “specificity” 0.61 or 0.76? It can be clearly explained.

10.  In Figure 3, the texts within the figure are unclear, please improve the resolution.

11.  In the second and third paragraphs of page 21, are the orders of the cited references wrong? The references 35-42 and 67-98 are described, however, the number of cited references has reached [224] in this page.

Author Response

  1. In the author list, it seems to be incomplete in the end: “…Akila Prashant a and c*”. - We thank the reviewer for identifying the error which has been rectified in the revised manuscript
  2. The line numbers are lacking in this manuscript. - line numbers are added in the revised manuscript. 
  3. In the Introduction section, the specific aims can be added. - As suggested by the reviewers specific aim is added in the introduction section. 
  4. The format of this manuscript can be modified a little. Too many square brackets ([ ]) are used in the manuscript, which can easily be confused with cited references. It is recommended to use round brackets (( )) except for square brackets for cited references. - As suggested by the reviewer round brackets are used throughout the revised manuscript. 
  5. In Figure 1, the presentation is confusing. The figure legend is too brief to understand the meaning of the figure. What are the meanings of (a), (b), and (c)? What does it mean for the dot-lines in the middle of the figure? - Figure 1 is modified and the legend is rewritten as suggested by the reviewers. 
  6. The descriptions in the second line from the bottom, page 4 to line 27, page 5 are confusing. It should be re-organized and clearly explained. - This section has been modified and re-organised as suggested by the reviewer. 
  7. In Figure 2, the whole names for abbreviations are recommended to be described in the figure legend. - The legend and abbreviations in figure 2 have been modified. 
  8. In page 9, line 2, [> 12 × 109/L] can be changed to [> 1.2 × 1010/L]. - We have modified this as suggested by the reviewer. 
  9. In the middle paragraph of page nine, what are the meanings for “sensitivity” 0.8 or 0.55 and “specificity” 0.61 or 0.76? It can be clearly explained. - We have addressed this error in the revised manuscript.
  10. In Figure 3, the texts within the figure are unclear; please improve the resolution. - The resolution of Figure 3 has been improved, as suggested by the reviewer. 
  11. In the second and third paragraphs of page 21, are the orders of the cited references wrong? The references 35-42 and 67-98 are described, however, the number of cited references has reached [224] in this page. - The numbering of the references have been checked in the revised manuscript. 

Round 2

Reviewer 3 Report

Comments and Suggestions for Authors

This revised manuscript has a great improvement and can be accepted.